# Glial-dependent clustering of voltage-gated ion channels in *Drosophila* precedes myelin formation

**Simone Rey[1†], Henrike Ohm[1†], Frederieke Moschref[1‡], Dagmar Zeuschner[2], Marit Praetz[1], Christian Klämbt[1]\***

[1]Institut für Neuro- und Verhaltensbiologie; Röntgenstraße, Münster, Germany; [2]Max-Planck Institut für molekulare Biomedizin; Wissenschaftliches Service-Labor für Elektronenmikroskopie; Röntgenstraße, Muenster, Germany

**Abstract** Neuronal information conductance often involves the transmission of action potentials. The spreading of action potentials along the axonal process of a neuron is based on three physical parameters: the axial resistance of the axon, the axonal insulation by glial membranes, and the positioning of voltage-gated ion channels. In vertebrates, myelin and channel clustering allow fast saltatory conductance. Here, we show that in *Drosophila melanogaster* voltage-gated sodium and potassium channels, Para and Shal, co-localize and cluster in an area resembling the axon initial segment. The local enrichment of Para but not of Shal localization depends on the presence of peripheral wrapping glial cells. In larvae, relatively low levels of Para channels are needed to allow proper signal transduction and nerves are simply wrapped by glial cells. In adults, the concentration of Para increases and is prominently found at the axon initial segment of motor neurons. Concomitantly, these axon domains are covered by a mesh of glial processes forming a lacunar structure that possibly serves as an ion reservoir. Directly flanking this domain glial processes forming the lacunar area appear to collapse and closely apposed stacks of glial cell processes can be detected, resembling a myelin-like insulation. Thus, *Drosophila* development may reflect the evolution of myelin which forms in response to increased levels of clustered voltage-gated ion channels.

**\*For correspondence:**
klaembt@uni-muenster.de

[†]These authors contributed equally to this work

**Present address:** [‡]Max-Planck Institut für Multidisziplinäre Naturwissenschaften; Hermann Rein-Str, Göttingen, Germany

**Competing interest:** The authors declare that no competing interests exist.

## Editor's evaluation

Here, the authors characterize axon-initial segment-like structures in *Drosophila* using a variety of approaches spanning molecular genetic, confocal and super-resolution imaging, and ultrastructure. These important findings advance our understanding of how myelin evolved with potentially early roles in organizing ion channel clustering in axons. The convincing evidence supporting the conclusions includes advanced imaging combined with molecular genetic approaches, which will be of significant interest to neuroscientists and cell biologists.

## Introduction

A functional nervous system requires the processing and transmission of information in the form of changing membrane potentials. To convey information along axons, neurons generate action potentials by opening of evolutionarily conserved voltage-gated sodium and potassium channels (*Moran et al., 2015*). Once an action potential is generated, it travels toward the synapse and the speed of information transfer is of obvious importance. It is long established that axonal conductance velocity depends on the resistance within the axon, which inversely correlates with its diameter. In addition, it depends on the resistance across the axonal membrane, which is increased by extensive

glial wrapping. Furthermore, spacing of voltage-gated ion channels contributes to axonal conduction velocity (*Eshed-Eisenbach and Peles, 2021*; *Freeman et al., 2016*; *Hodgkin and Huxley, 1952*).

In vertebrates, unmyelinated axons generally have a small diameter with evenly distributed voltage-gated ion channels along their plasma membrane, and in consequence their conductance velocity is slow (*Castelfranco and Hartline, 2015*). To speed up conductance, axons grow to a larger diameter and show a clustering of voltage-gated ion channels at the axon initial segment (AIS) and the nodes of Ranvier. Together with the insulating glial-derived myelin sheet, this allows fast saltatory conductance (*Arancibia-Cárcamo et al., 2017*; *Castelfranco and Hartline, 2015*; *Cohen et al., 2020*; *Dutta et al., 2018*; *Eshed-Eisenbach and Peles, 2021*).

In invertebrates, mechanisms to increase conductance speed are thought to be limited by radial axonal growth, as seen in the giant fiber system of *Drosophila* or the giant axon of the squid (*Allen et al., 2006*; *Hartline and Colman, 2007*). No saltatory conductance has been described for invertebrates and it is assumed that voltage-gated ion channels distribute relatively evenly along axonal membranes. Nevertheless, myelin-like structures were found in several invertebrate species, including annelids, crustacean, and insects (*Coggeshall and Fawcett, 1964*; *Davis et al., 1999*; *Günther, 1976*; *Hama, 1959*; *Hama, 1966*; *Hess, 1958*; *Heuser and Doggenweiler, 1966*; *Levi et al., 1966*; *Roots, 2008*; *Roots and Lane, 1983*; *Wigglesworth, 1959*; *Wilson and Hartline, 2011*). However, it is unknown whether such myelin-like structures also impact the distribution of ion channels.

To address how glial cells affect axonal conductance velocity we turned to *Drosophila*. In the larvae, peripheral axons are engulfed by a single glial wrap resembling Remak fibers in the mammalian PNS (*Matzat et al., 2015*; *Nave and Werner, 2014*; *Stork et al., 2008*). In addition to insulating axons, we found that glial cells promote radial axonal growth. In the absence of wrapping glia axons are not only thin, but they are also characterized by a severe reduction in conductance velocity, which is stronger than predicted by the reduced axonal diameter (*Hodgkin and Huxley, 1952*; *Kottmeier et al., 2020*). Thus, wrapping glial cells might control localization of voltage-gated ion channels along the axonal plasma membrane.

## Results

### Distribution of the voltage-gated sodium channel Para

The *Drosophila* genome harbors only one voltage-gated sodium channel called Paralytic (Para), which is required for the generation of all action potentials (*Kroll et al., 2015*). To study the localization of Para and to test whether *Drosophila* glia affects its localization, we and others tagged the endogenous *para* locus with all predicted isoforms being modified (*Ravenscroft et al., 2020*; *Venken et al., 2011*; *Figure 1A*). In *para^mCherry* flies, monomeric Cherry (mCherry) is inserted close to the Para N-terminus (*Figure 1A*). Homozygous or hemizygous *para^mCherry* flies are viable with only mildly affected channel function (*Figure 1B*; *Ravenscroft et al., 2020*; *Venken et al., 2011*). Para^mCherry localizes along many CNS and PNS axons of the larval nervous system (*Figure 1—figure supplement 1A–C*; *Ravenscroft et al., 2020*; *Venken et al., 2011*).

To independently assay Para localization, we generated antibodies against an N-terminal epitope shared by all predicted Para isoforms (*Figure 1A*). In western blots, anti-Para antibodies detect a band of the expected size (>250 kDa), which is shifted toward a higher molecular weight in protein extracts of homozygous *para^mCherry* animals (*Figure 1—figure supplement 1D*). Immunohistochemistry detects Para localization in control first instar larvae but not in age-matched *para* null mutant animals, further validating the specificity of the antibodies (*Figure 1D–E'*). Whereas the pre-immune serum fails to detect any specific proteins (*Figure 1F*), anti-Para antibody staining of third instar larval filets revealed the localization of Para in the CNS and the PNS (*Figure 1G*) similar to what was noted for Para^mCherry localization (*Figure 1C*). Thus, we anticipate that endogenously mCherry-tagged Para protein reflects the wild typic Para localization.

To test a possible differential distribution of Para in either sensory or motor axons, we utilized RNAi to remove *mCherry* expression in heterozygous *para^mCherry* females. This leaves the wild type *para* allele intact and circumvents the early lethal phenotype associated with loss of *para*. Knockdown of *mCherry* expression in glutamatergic motor neurons (*Mahr and Aberle, 2006*) reveals *para^mCherry* expression in cholinergic sensory neurons of third instar larvae. Here, Para appears to evenly localize along the abdominal nerves and is found at many processes within the CNS (*Figure 2A*). In contrast, silencing *para^mCherry* in cholinergic neurons (*Salvaterra and Kitamoto, 2001*) reveals a predominant localization of Para in an axonal segment of motor axons at the PNS/CNS boundary of third instar larvae (*Figure 2B*), as suggested before (*Ravenscroft et al., 2020*).

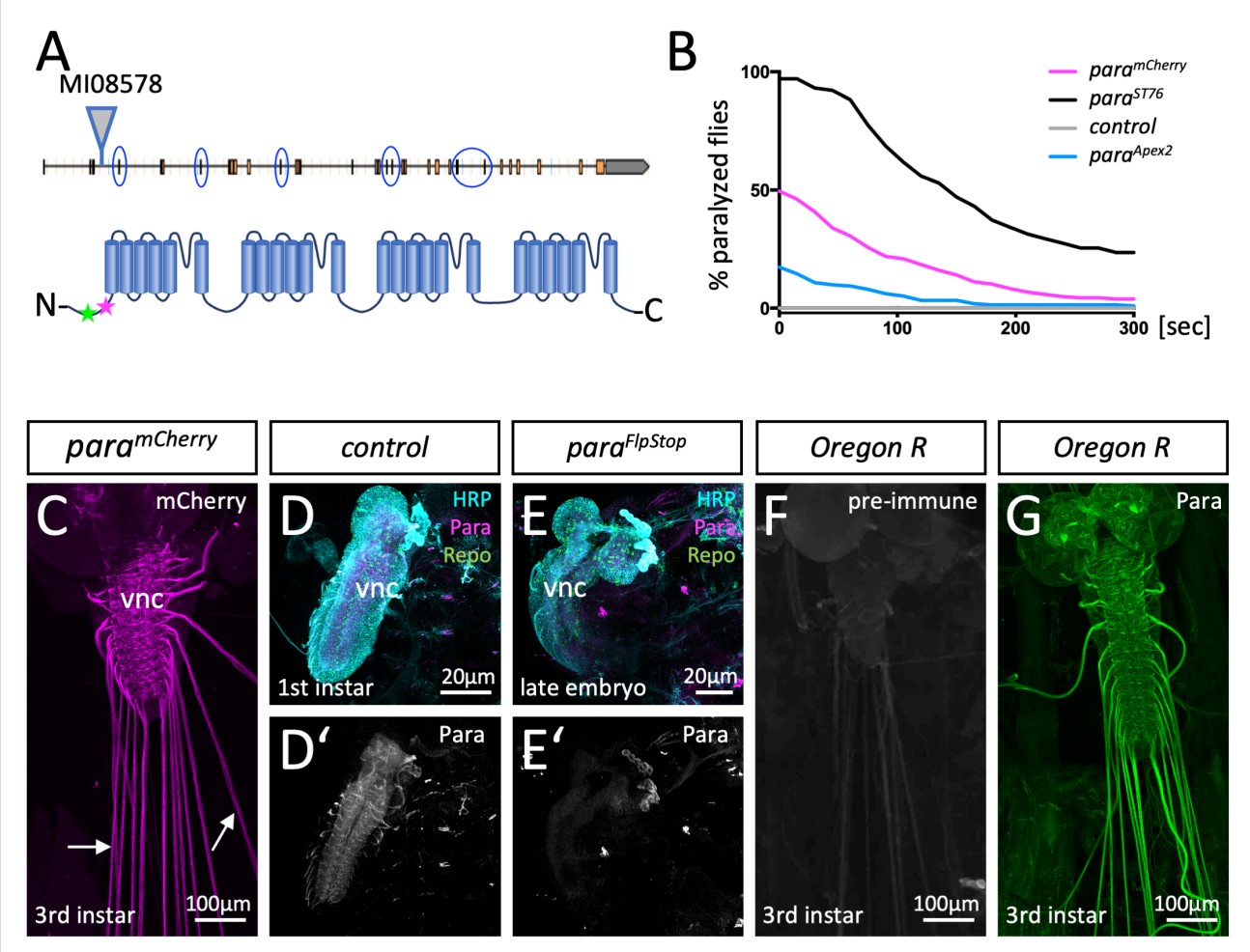

**Figure 1.** Localization of Paralytic (Para) voltage-gated ion channels in the larval nervous system. (**A**) Schematic view on the *para* gene. Alternative splicing at the circled exons results in the generation of more than 60 Para isoforms. All isoforms share a common N-terminus. Here, the MiMIC insertion *MI08578* allows tagging of the endogenous *para* gene. The peptide sequence AEHEKQKELERKRAEGE (positions 33–49) that was used for immunization is indicated by a green star, the MiMIC insertion is indicated by a magenta star. (**B**) Homozygous *para^mCherry^*, *para^Apex2^* or *para^ST76^* flies were tested for temperature-induced paralysis. The recovery time is indicated. (**C**) Third instar larval *para^mCherry^* nervous system stained for Cherry localization. Para^mCherry^ is detected in the ventral nerve cord (vnc) and diffusely along peripheral nerves (arrows). (**D,D'**) Affinity purified anti-Para antibodies detect a protein in the CNS of dissected 24-hr-old wild type first instar larvae. (**E,E'**) No protein is found in the CNS of dissected age-matched *para* mutant animals. (**F**) Third instar larval nervous system stained with the pre-immune control. (**G**) Third instar larval nervous system stained with affinity purified anti-Para antibodies. Scale bars are as indicated.

The online version of this article includes the following source data and figure supplement(s) for figure 1:

**Figure supplement 1.** Ensheathing- / wrapping glia in Drosophila and characterization of Para protein.

**Figure supplement 1—source data 1.** Four images of western blots, with and without marker bands, are provided.

## Differential localization of voltage-gated potassium channels

Several genes encode voltage-gated potassium channels. Using endogenously tagged proteins, we find the Shaker potassium channel mostly in the synaptic neuropil regions (*Figure 2C*). The distribution of Shab resembles Para^mCherry^ localization in sensory axons (*Figure 2A and D*), whereas Shal localizes in a pattern similar to Para localization on motor axons (*Figure 2B and E*). Thus, *Drosophila* larval motor axons appear to have an axonal segment resembling the vertebrate AIS harboring both voltage-gated sodium and potassium channels.

We then analyzed the localization of Para in the adult CNS. Here, too, global Para localization, as detected by anti-Para antibodies, matched the Para^mCherry^ signal (*Figure 2F and G*). To differentiate

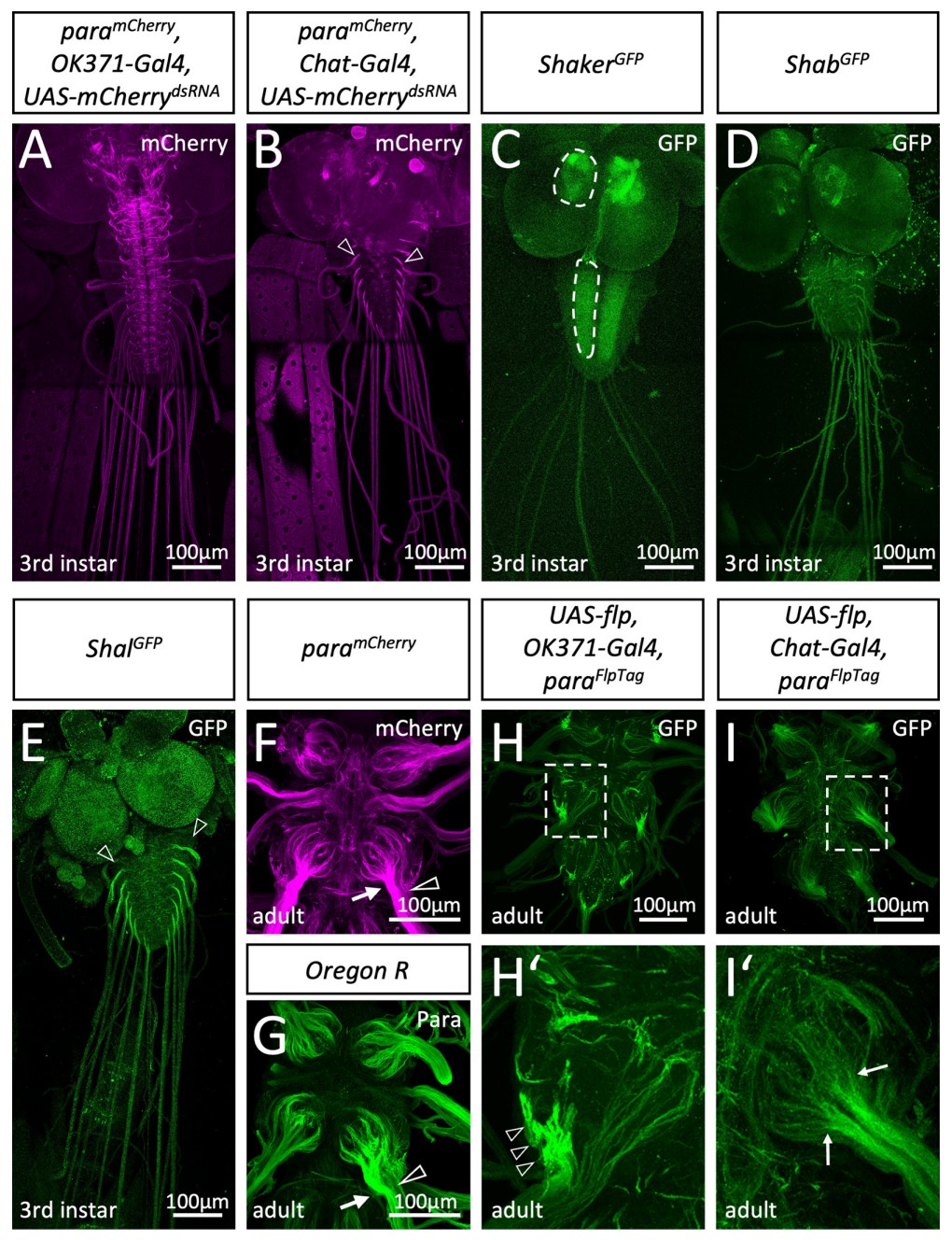

**Figure 2.** Differential localization of voltage-gated ion channels in *Drosophila*. (**A**) Third instar larvae with the genotype [*para^mCherry^; OK371-Gal4, UAS-mCherry^dsRNA^*]. *para^mCherry^* expression is suppressed in all glutamatergic neurons and thus, Para^mCherry^ localization along axons of cholinergic sensory neurons becomes visible. (**B**) Third instar larvae with the genotype [*para^mCherry^; Chat-Gal4, UAS-mCherry^dsRNA^*]. Here, expression of *para^mCherry^* is suppressed in all cholinergic neurons which reveals Paralytic (Para) localization in motor neurons. Note the prominent Para localization at the CNS/PNS transition point (arrowheads). (**C**) Third instar larval *shaker^GFP^* nervous system stained for GFP localization. Shaker is found in the neuropil (dashed areas). (**D**) Third instar larval *shab^GFP^* nervous system stained for GFP localization. Shab is distributed evenly along all peripheral axons. (**E**) Third instar larval *shal^GFP^* nervous system stained for GFP localization. Shal localizes similar as Para on motor axons. Scale bars are 100 μm. (**F**) Adult *para^mCherry^* ventral nerve cord stained for Para localization. Para^mCherry^ localizes prominently along segments of peripheral nerves (arrow) as they enter thoracic neuromeres. Note that some axons entering the CNS neuropil show only a weak Para signal (open arrowhead). (**G**) Control (*Oregon R*) adult ventral nerve cord stained for Para protein localization using purified anti-Para antibodies. Note the differential localization

*Figure 2 continued on next page*

*Figure 2 continued*

of Para along axons entering the nerve (arrow, open arrowhead). (**H**) Ventral nerve cord of an adult fly with the genotype [*para^FlpTag-GFP*; *Ok371-Gal4, UAS-flp*]. The boxed area is shown enlarged in (**H′**). The arrowheads point to high density of Para. (**I**) Ventral nerve cord of an adult fly with the genotype [*para^FlpTag-GFP*; *Chat-Gal4, UAS-flp*]. The boxed area is shown enlarged in (**H′**). Note that Para localization is reduced as soon as axons enter the neuropil (arrows). Scale bars are as indicated.

between Para expression in motor and sensory neurons, we employed the recently developed FlpTag technique (*Fendl et al., 2020*). Here, cell type-specific expression of the Flp recombinase induces the inversion of a GFP-encoding exon located in the gene of interest. In *para^FlpTag* flies (*Fendl et al., 2020*), Flp expression in all motor neurons results in strong labeling of Para localization at a small part of the axon as it leaves the neuropil (*Figure 2H and H′*, arrowheads), indicating that an AIS is also found in adult motor axons. In contrast, expression of Flp in all sensory neurons using *Chat-Gal4 UAS-flp* reveals an even Para decoration of axons as they enter the CNS which fades out when axons reach into the neuropil (*Figure 2I,I′*, arrows).

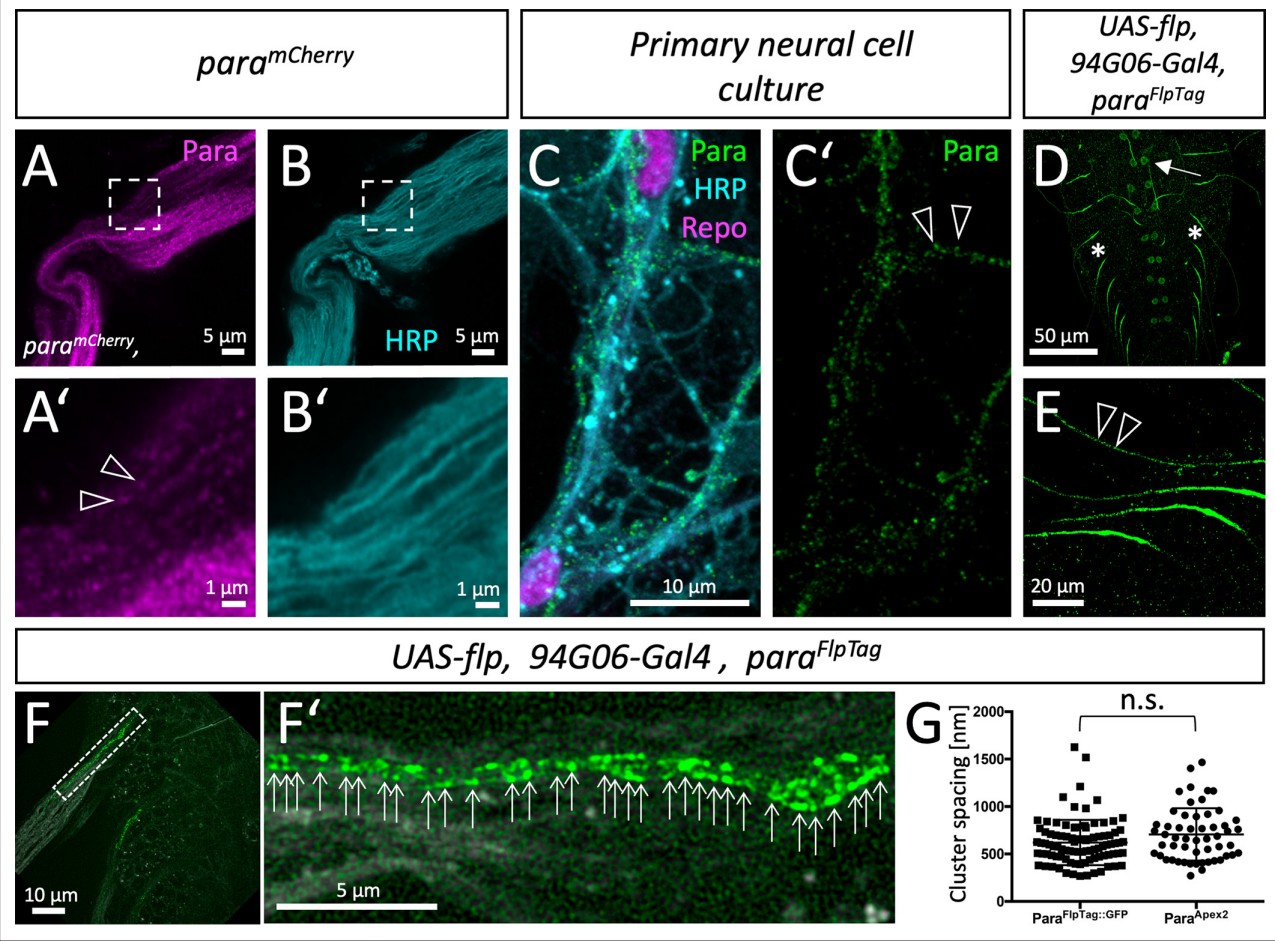

**Figure 3.** Clustered localization of Paralytic (Para) along motor axons. (**A**) High-resolution Airyscan analysis of Para^mCherry and (**B**) HRP localization in an adult nerve. The boxed area is shown in higher magnification below (**A′,B′**). Note the clustered appearance of Para^mCherry, clusters are about 0.6–0.8 μm apart (arrowheads). (**C,C′**) Primary wild type neural cells cultured for 7 days stained for Repo (magenta) to label glial nuclei, HRP (cyan) to label neuronal cell membranes and anti-Para antibodies (green). The Para protein localizes in a dotted fashion. (**D**) Ventral nerve cord of a third instar larva with the genotype [*para^FlpTag-GFP*; *94G06-Gal4, UAS-flp*]. The arrow points to a single neuronal cell body found in every hemineuromer. (**E**) Higher magnification of single Para^GFP expressing axons. Note the dotted arrangement of Para^GFP along motor axons (arrowheads). (**F,F′**) Ventral nerve cord of a third instar larva with the genotype [*para^FlpTag-GFP*; *94G06-Gal4, UAS-flp*] imaged with super-resolution. The dashed box is shown in high magnification in (**F′**). Arrows point to clusters of Para protein. Scale bars are as indicated. (**G**) Quantification of Para cluster distance using super-resolution imaging (Para^FlpTag::GFP, average distance is 620 nm, n=91 clusters on 3 axons, 2 larvae) or electron microscopy (Para^Apex2, average distance is 706 nm, n=64 clusters on 8 axons 4 animals, Mann-Whitney U-test, p=0.0747, two-tailed). Scale bars are as indicated.

## High-resolution imaging reveals clustered localization of Para along motor axons

To obtain a higher spatial resolution of Para distribution, we used high-resolution Airyscan microscopy. In adult nerves, Para$^{mCherry}$ localization distal to the AIS is found in a clustered arrangement (*Figure 3A–B'*, arrowheads). To exclude that cluster formation is due to the fluorescence protein moiety, we performed anti-Para immunohistochemistry on primary *Drosophila* neural cells in culture, where axons form small fascicles with few accompanying glial cells (*Figure 3C and C'*). When such cultures are stained for Para distribution, we find Para channels localized in small clusters with a spacing of about 0.6–0.8 μm. However, in these neuronal cultures we cannot clearly define the number of Para expressing axons in a fascicle.

To further improve spatial resolution, we combined high-resolution imaging with the FlpTag labeling method. We restricted Flp expression to only one motor neuron in each larval hemineuromer using *94G06-Gal4* (*Jenett et al., 2012*; *Pérez-Moreno and O'Kane, 2019*), which results in the expression of GFP-tagged Para channels in only a single neuron. Whereas weak expression is noted around the nucleus, strong expression is seen in the AIS (*Figure 3D*, arrow, asterisks). Flanking the strong expression along the AIS, a clustered localization of Para$^{GFP}$ could be noted (*Figure 3E*, arrowheads). Further super-resolution imaging of single motor axons decorated with GFP-tagged Para showed an average spacing of 0.620 μm (*Figure 3F, F', and G*, n=91 clusters on 3 axons, quantification using Fiji). Interestingly, Para clusters appear to be organized along lines at the motor axon which was also found when analyzing the distribution of Para at the electron microscopic level (see below).

## In sensory neurons increased Para localization is found at dendrites and the AIS

Having shown that in motor axons Para is concentrated in a clustered arrangement in an AIS of motor axons, we wondered whether similar distribution can be found in sensory axons. For this we expressed *flp* in multidendritic sensory neurons using the *pickpocket* Gal4 driver (*ppk-Gal4*). This allows labeling of the v'ada neurons (*Figure 4A*). Low levels of Para protein localize to the cell body and the distal shaft of the axon. Along the descending axon, Para localization increases only in some distance to the soma (*Figure 4A, B, and C*). However, *para* expression in sensory neurons is not as strong as in motor axons which may correspond to the notion that sensory axons are usually smaller axons. The relatively low expression levels did not allow super-resolution imaging and thus, we could not address whether Para is found in a clustered organization along axons of *ppk* positive sensory neurons. Interestingly, however, within some of the v'ada dendrites, Para accumulates in distinct clusters (*Figure 4A' and B'*, arrows).

In conclusion, the above data show the presence of an AIS in *Drosophila* motor and sensory axons. In motor axons, where this domain likely serves as a spike initiation zone (*Günay et al., 2015*), Para channels are organized in a clustered arrangement.

## Electron microscopic analysis of Para cluster formation

To determine the distribution of Para on the subcellular level, we integrated an Apex2 encoding exon in the *para* locus, which allows generating local osmiophilic diaminobenzidine (DAB) precipitates that are detectable in the electron microscope (*Lam et al., 2015*). The insertion of an Apex2 encoding exon in the N-terminus of Para affected *para* function less strongly than the insertion of an *mCherry* exon and resulted in a very weak hypomorphic *para* allele (*Figure 1A and B*). In adult flies, Para$^{Apex2}$ is expressed in sufficient intensity to be detected along axons using the electron microscope. In small diameter peripheral axonal segments, weak Para$^{Apex2}$ directed DAB precipitates are found (*Figure 5A*, white arrowheads). In contrast, in large diameter axons next to the CNS/PNS boundary intense DAB precipitates can be detected (*Figure 5B*). When we performed serial sectioning, the intensity of DAB labeling varied (*Figure 5B–D*), possibly reflecting the clustered localization of Para that we had found using the confocal microscope. We next determined the distribution of DAB precipitates along the circumference of an axon over 16 consecutive cross sections (*Figure 5E and F*). The resulting surface plot shows Para$^{Apex2}$ localization of a small segment of the axon in a 3D space. This suggests that Para$^{Apex2}$ clusters are organized in two lines along the ≈2.4 μm axonal circumference (*Figure 5F and G*), resembling the distribution of Para$^{mCherry}$ clusters along lines as detected using super-resolution light microscopy (*Figure 3F*). To further address the spacing of Para$^{Apex2}$ clusters along the longitudinal axis of the axon, we performed longitudinal sections of Para-rich axon segments and determined the staining intensity along the plasma membrane by using Fiji (*Figure 5H,I*). Here, again a spatial modulation of the Para staining intensity is apparent, with a spacing of 0.706 μm (*Figure 5I*; see *Figure 3G*

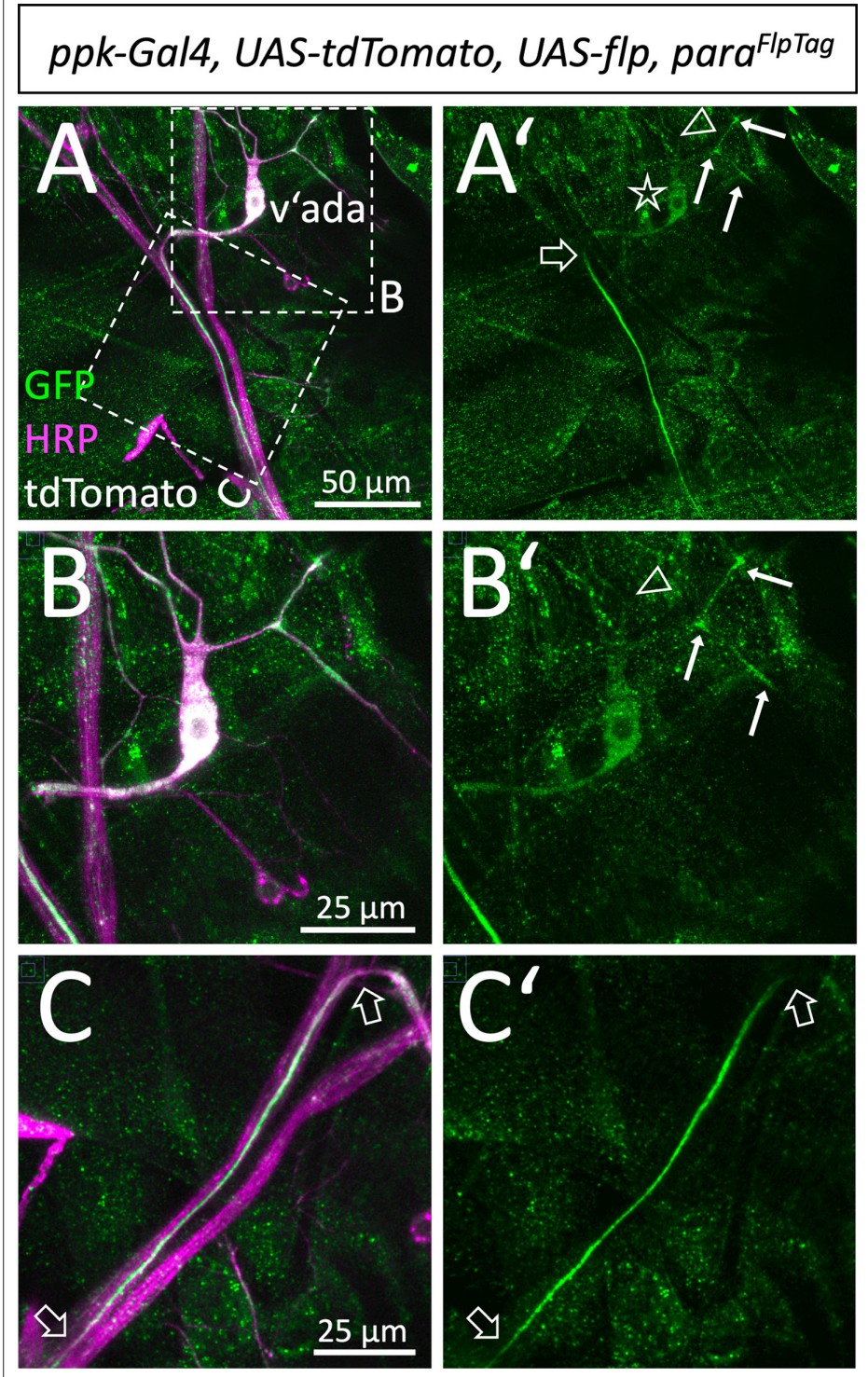

**Figure 4.** Localization of Paralytic (Para) along sensory axons. (**A,A′**) Ventral *pickpocket* expressing sensory neuron (v′ada) of a third instar larva with the genotype [*para^FlpTag-GFP*; *ppk-Gal4, UAS-flp, UAS-tdTomato*] stained for GFP (green), HRP (magenta), and tdTomato (white). The dashed boxes are shown in higher magnification in (**B,C**). The asterisk denotes the position of the neuronal cell soma. The filled arrows indicate localized Para along some of the dendritic processes. The open arrowhead points to a dendritic process lacking Para localization. Note that Para localization along the descending axon becomes prominent only after about 50 μm (open arrow). (**B,B′**) Magnification of the neuronal soma attached dendrites. (**C,C′**) Descending axon of the v′ada neuron. Note that the strong Para signal starts 50 μm distal to the cell soma and fades out after 100 μm (open arrows). Scale bars are as indicated.

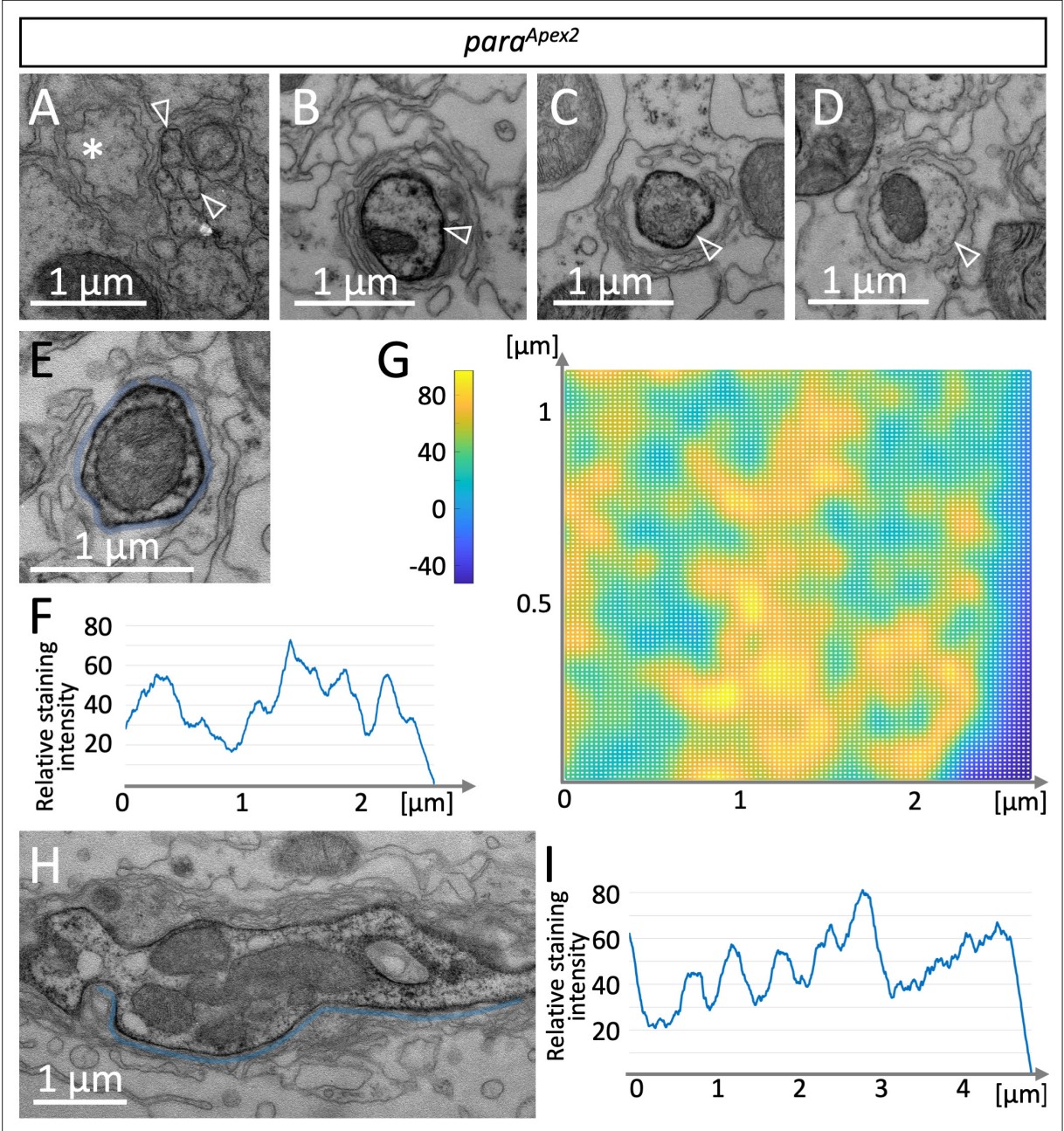

**Figure 5.** A glial lacunar system surrounds the axon initial segment. (**A**) Weak Paralytic (Para) expression can be detected on *para*^Apex2^ expressing small axons (arrowheads) running in fascicles within the nerve. (**B–D**) Cross sections through the same axon at various positions. Distance between individual sections (**B,C**) is 15 µm, distance between (**C,D**) is 6.5 µm. Note the intense labeling of the axonal membrane is changing between the different sections. (**E**) Cross section, to determine the staining intensity along the membrane (below the blue line), a corresponding ROI was defined and (**F**) quantified using Fiji. (**G**) Surface plot of Para^Apex2^ distribution along 16 consecutive axonal cross sections. For details, see Materials and methods. The intensity of diaminobenzidine (DAB) precipitates is transformed to different colors. Note that Para clusters are organized in two longitudinal lines across the axonal membrane surface. (**H**) Longitudinal section of a *para*^Apex2^ expressing axon. The staining intensity along the membrane (above the blue line) was quantified using Fiji. (**I**) Staining intensity of the membrane stretch shown in (**H**). Note the regular increase in staining intensity every 0.6–0.8 µm. For quantification see *Figure 3G*. Scale bars are as indicated.

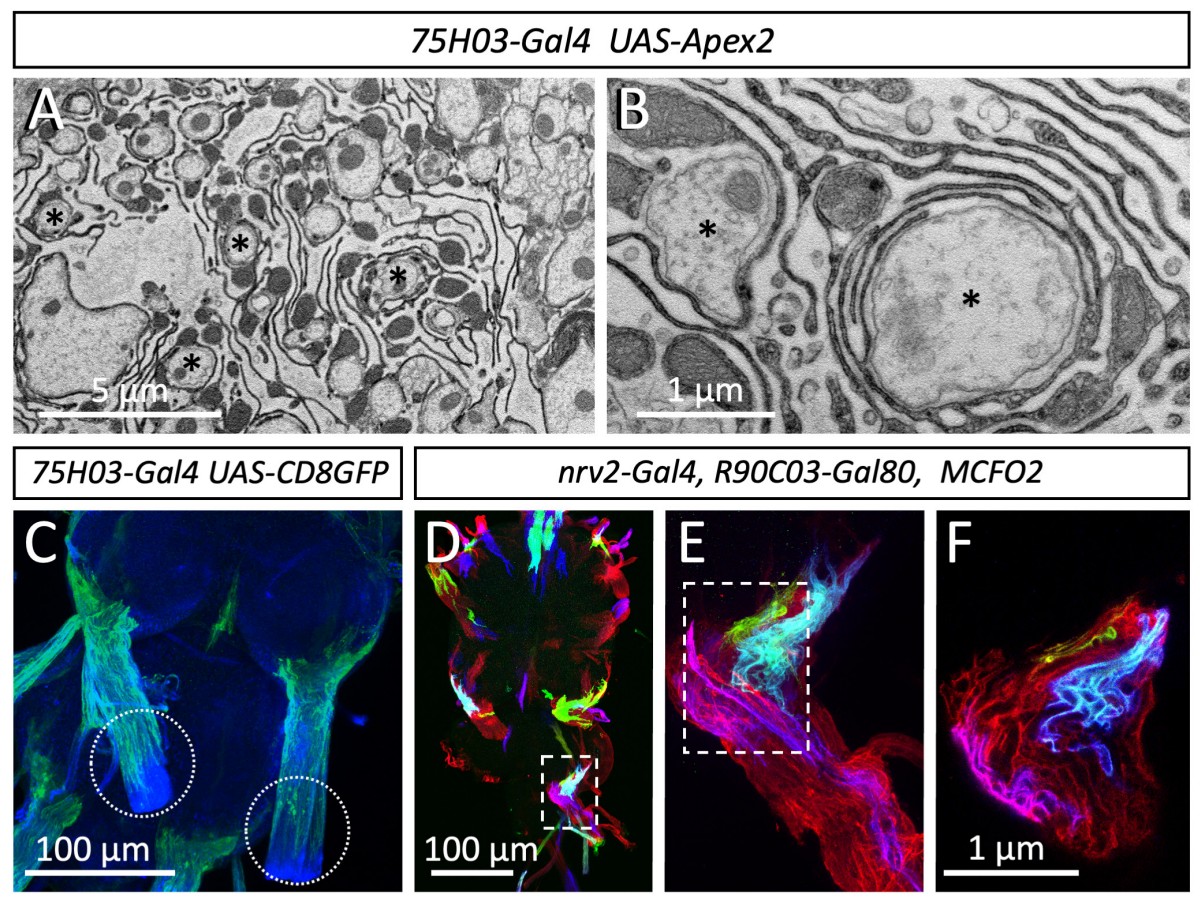

**Figure 6.** Organization of the lacuna forming tract glial. (**A,B**) Apex2 expression directed by *75H03-Gal4*. Axons (asterisks) are engulfed by lacunar structures that are largely formed by the tract glia. (**C**) Maximum projection of a confocal image stack. *75H03-Gal4* directed expression of GFP labels the ensheathing/wrapping or tract glia. Note that GFP expression ends proximal to the dissection cut (white dashed circles). (**D–F**) multicolor flipout (MCFO2) analysis of the *nrv2-Gal4, R90C03-Gal80* positive wrapping glia. Note that glial cells tile the nerve roots with no gaps in between. Scale bars are as indicated.

The online version of this article includes the following figure supplement(s) for figure 6:

**Figure supplement 1.** Glial cell types involved in lacuna formation.

for quantification, n=54 cluster distances on 6 axons), which is similar to what we determined by confocal microscopy.

## Para-rich axon segments are embedded in a lacunar system formed by tract glia

Interestingly, the large caliber axons decorated with highest levels of Para protein are embedded in a mesh-like glial organization, that resembles the lacunar system described earlier for the cockroach (*Figure 5B–D*, *Figure 6A, B*; *Wigglesworth, 1960*). The *Drosophila* glial lacunar system is characterized by intensive formation of glial processes around axons which are always larger than 0.5 µm in diameter (*Figure 6A, B*, asterisks). Glial cell processes have an average thickness of 35 nm (*Figure 6A, B*, n=189 processes, 4 nerves from 3 animals).

Next, we determined which glial cell type forms these lacunar structures. In the larva, the central ensheathing/wrapping glial cells express the *83E12-Gal4* driver (*Peco et al., 2016*; *Pogodalla et al., 2021*), whereas the peripheral wrapping glial cells can be addressed using the *nrv2-Gal4 90C03-Gal80* driver (*Kottmeier et al., 2020*; *Matzat et al., 2015*; *Stork et al., 2008*; *Figure 1—figure supplement 1A and B*). In adults – but not in larvae – a specialized group of glial cells is found at the CNS/PNS boundary, called tract glia (*Kremer et al., 2017*) which overlaps with both the central ensheathing glia

and the peripheral wrapping glia (*Figure 6—figure supplement 1*). Interestingly, a similarly distinct group of glial cells has been identified in the vertebrate nervous system (*Fontenas and Kucenas, 2017*; *Kucenas et al., 2008*; *Kucenas et al., 2009*). The position of the lacunae coincides with the location of the tract glial cells (*Kremer et al., 2017*) (compare *Figures 2H, 6C*). These glial cells express *75H03-Gal4*, *83E12-Gal4* as well as the *nrv2-Gal4 90C03-Gal80* driver (*Figure 6—figure supplement 1*). Multicolor flipout (MCFO2) labeling experiments (*Nern et al., 2015*) indicate tiling of these glial cells along the nerve with no overlap and no spaces in between individual glial cells (*Figure 6D–F*). To further determine which glial cell forms the lacunar structures, we generated flies harboring a *UAS-Myr-Flag-Apex2-NES* transgene (Apex2$^{Myr}$, see Materials and methods) and expressed the myristoylated Apex2 with the different Gal4 drivers mentioned above. These experiments confirmed that most of the lacunar system is indeed generated by tract glial cell processes (*Figure 6A, B*).

Thus, in large caliber motor axons, most of the Para voltage-gated sodium channels is positioned close to the lacunar system, which had been previously speculated to serve as an extracellular ion reservoir needed for sustained generation of action potentials (*Chandra and Singh, 1983*; *Leech and Swales, 1987*; *Maddrell and Treherne, 1967*; *Treherne and Schofield, 1981*; *Van Harreveld et al., 1969*; *Wigglesworth, 1960*).

## Myelin in the leg nerve is found close to the CNS

In vertebrates, clustering of voltage-gated ion channels occurs on the edges of myelinated axonal segments (internodes) (*Arancibia-Cárcamo et al., 2017*; *Castelfranco and Hartline, 2015*; *Cohen et al., 2020*; *Dutta et al., 2018*; *Eshed-Eisenbach and Peles, 2021*). Here, myelin not only participates in positioning of voltage-gated ion channels but also increases electric insulation and thus contributes to a faster conductance velocity. In *Drosophila* highest conductance velocity is likely to be required during fast and well-tuned locomotion in adults. Thus, we focused our search for myelin-like structures on adult leg nerves. Most of the 760 axons within an adult leg run in a single large nerve that exit the CNS at well-defined positions (*Figure 7A-C*, *Figure 7—figure supplement 1A*). Unlike the organization in larval nerves, axons running in the leg nerves are found in distinct zones depending on their diameter (*Figure 7B and C*). At the position of the femur, large axons are always covered by a single glial sheet. Small diameter axons are generally not individually wrapped but rather engulfed as a fascicle (*Figure 7B and D*). At the coxa, close to the CNS, we noted that large diameter axons were occasionally flanked by several glial membrane sheets (*Figure 7E*, asterisk). Up to 15 flat glial membrane sheets with a thickness of about 28 nm are found along larger axons (*Figure 7E,F*, *Figure 7—figure supplement 1B*). Axons with an intermediate diameter show individual glial wrapping with a single or very few glial sheets (*Figure 7E and H*). To quantify the occurrence of myelin-like structures we made semi-serial distal to proximal sections of six nerves every 5 µm across the entire lacunar area spanning 40–60 µm (*Figure 7—figure supplement 2*). The position where lacunar structures were first identified was set as zero. We then counted the occurrence of myelin-like structures in every section that we defined as ≥4 glial layers in close apposition. Here, we noted an increase in the number of myelin-like structures at the distal end of the lacunae (*Figure 7—figure supplement 2B*, *Figure 7—figure supplement 3*). No myelin-like structures were found at proximal positions close to the neuropil. The position of the up to four myelin-like structures found within a section plane was variable and could be either at the margin of the lacunae (*Figure 7—figure supplement 3A*) or could be found separating an area with small axons from an area with large axons (*Figure 7—figure supplement 3B*), or close to the blood-brain barrier (*Figure 7—figure supplement 3C*). In rare cases we noted formation of myelin-like membrane stacks without contact to axons in the lacunar region (*Figure 7—figure supplement 3D*). Myelin-like sheets contact serval axons (*Figure 7—figure supplement 3A–C, E and F*) but can also engulf single large axons with varying complexity of the membrane stacks (*Figure 7—figure supplement 3G–H*, *Figure 7—figure supplement 4*).

## Myelin can be formed by central tract glia and peripheral wrapping glia

To determine which glial cell type is able to form myelin-like structures, we expressed Apex2$^{Myr}$ in specific glial cell types and analyzed whether DAB positive myelin-like stacks of glial cell processes can be detected in the electron microscope. Upon expression of Apex2$^{Myr}$ in CNS-derived tract glia *75H03-Gal4* DAB positive myelin stacks can be detected (*Figure 7G* (black arrowhead), *Figure 7I*, *Figure 7—figure supplement 3*, *Figure 7—figure supplement 5*). The finding that *75H03-Gal4* directed Apex2 labeling can be found next to unlabeled glial sheets (*Figure 7G*, white arrowhead) suggests that peripheral wrapping glial cells can also form myelin-like structures in the leg nerve. *Drosophila* myelin-like membrane stacks are generated by extensive membrane folding providing the

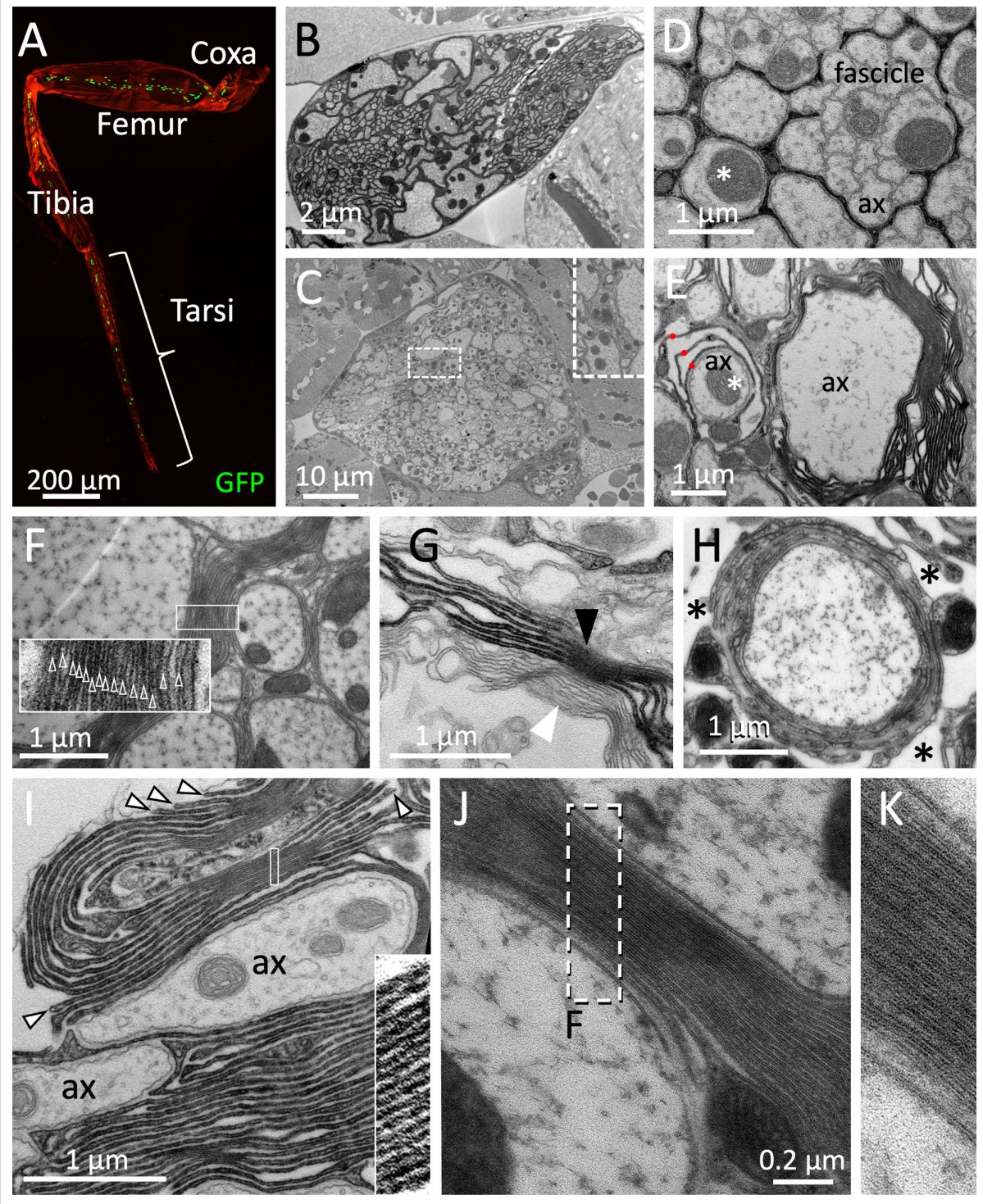

**Figure 7.** *Drosophila* wrapping glia form myelin. (**A**) *Drosophila* leg of a 3-week-old fly with wrapping glial nuclei in green, the cuticle is stained by autofluorescence, the genotype is [*nrv2-Gal4, UAS-lamGFP*]. (**B–K**) Electron microscopic images of sections taken from 3-week-old female flies. (**B**) Section at the level of the femur. (**C**) Electron microscopic section at the level of the coxa. In some areas, an increased amount of glial membranes can be detected close to large caliber axons (box with white dashed lines, enlarged as an inlay). (**D,E,G**) Cross sections through a 2 weeks adult leg of a fly with the genotype [*75H03-Gal4, UAS-Myr-Flag-Apex2-NES*]. Glial cell processes are stained by the presence of Apex2 which generates an osmiophilic diaminobenzidine (DAB) precipitate. (**D**) Small caliber axons (ax) are engulfed by a single glial process as fascicle. Larger axons are individually wrapped (asterisk). (**E**) Large caliber axons are surrounded by glial membrane stacks. The asterisk denotes an axon engulfed by a few glial wraps (red dots). (**F**) Up to 15 densely packed membrane sheets are found (see inlay for enlargement). (**G**) Darkly stained tract glia membrane stacks (black arrowhead) can be

*Figure 7 continued on next page*

*Figure 7 continued*

found next to unlabeled membrane stacks (white arrowhead), suggesting that myelin-like structures can be derived from both, central and peripheral wrapping glial cells. (**H**) High-pressure freezing preparation showing a single axon covered by myelin-like membrane sheets in a lacunar area (asterisks). (**I**) Note the bulged appearance of the growing tip of the glial cell processes that form the myelin-like structures (arrowheads). The inlay shows a highly organized membrane stacking. (**J,K**) High-pressure freezing preparation of prefixed samples to reduce tissue preparation artifacts. Note the compact formation of membrane layers. The white dashed area is shown in (**K**). Scale bars are as indicated.

The online version of this article includes the following figure supplement(s) for figure 7:

**Figure supplement 1.** Count of axons in Drosophila leg nerves and measurement of glial cell process width.

**Figure supplement 2.** Extent of the lacunar system.

**Figure supplement 3.** Quantification of myelin distribution in the leg nerve.

**Figure supplement 4.** Multilayered myelin-like structures are formed around single axons in the adult nervous system.

**Figure supplement 5.** Formation of myelin-like structures in the adult CNS of *Drosophila*.

disadvantage that axons are not entirely insulated (*Figure 7I*, *Figure 7—figure supplements 4 and 5*). However, we occasionally do find axons encircled by multiple glial wraps (*Figure 7H*, *Figure 7—figure supplements 4 and 5*). Interestingly, in some areas we noted almost compacted glial membrane sheets (*Figure 7F,I*, inlay boxed areas).

To further validate these findings we performed additional high-pressure freezing of prefixed samples to optimize tissue preservation (*Möbius et al., 2016*; *Sosinsky et al., 2008*). In such specimens compact stackings of thin glial membrane sheets can be detected, too (*Figure 7J and K*). In the compacted areas (*Figure 7I–K*), the interperiodic distance of the different glial layers is about 30 nm, which is considerably more than the interperiodic distance of 13 nm found in mouse peripheral myelin (*Fledrich et al., 2018*). The unique compact appearance of vertebrate myelin is mediated by the myelin basic protein (MBP) (*Nave and Werner, 2021*). In contrast to vertebrate myelin where extra- and intercellular space is removed, fly myelin-like structures only show an irregular compaction of the extracellular space.

## Para localization depends on wrapping glial cells

Next, we wanted to test whether wrapping glial cells participate in the control of positioning voltage-gated ion channels. To address this, we ablated either peripheral wrapping glia or central ensheathing glia including the tract glia by directing the expression of the proapoptotic gene *hid* (*Kottmeier et al., 2020*; *Pogodalla et al., 2021*) and assayed the distribution of Shal and Para. Ablation of central or peripheral wrapping glial cells does not affect the distribution of the voltage-gated potassium channel Shal (*Figure 8—figure supplement 1*). Likewise, removal of the CNS-specific ensheathing glia does not affect Para localization in the larval nervous system (*Figure 8—figure supplement 2A–E*). In contrast, upon ablation of the peripheral wrapping glia a marked change in Para protein localization becomes obvious (*Figure 8—figure supplement 2A–B'*). In control larvae, anti-Para antibodies detect only a weak labeling of segmental nerves, but all nerves are intensely decorated with Para in wrapping glia ablated larvae. Whereas in wild type control larvae, 2.5 times more Para protein is found at the CNS/PNS transition zone compared to nerve segments on the muscle field, an almost even distribution is noted in glia ablated larvae (*Figure 8C*). In addition to the redistribution of Para protein along the axon, we also noted a twofold increase of *para* mRNA levels in further qRT-PCR experiments (*Figure 8D*).

Taken together, even in the small insect *Drosophila melanogaster*, myelin-like structures are formed (*Figure 9*). They are preferentially found distally to a lacunar region. The lacunae are formed by glial cell processes and comprise a large extracellular liquid filled space (*Figure 9*). Para voltage-gated sodium channels are differentially localized along sensory and motor neurons. In sensory neurons, Para expression is generally weaker and concentrates in an AIS but is also found in dendritic processes. In motor neurons Para localization is enriched in axonal segments that are running within the glial lacunar system. Interestingly, glia ablation experiments indicate that normal *para* mRNA expression as well as Para protein localization is dependent on the presence of wrapping glial cell processes. This suggests a signaling pathway from glia to the regulation of *para* transcription.

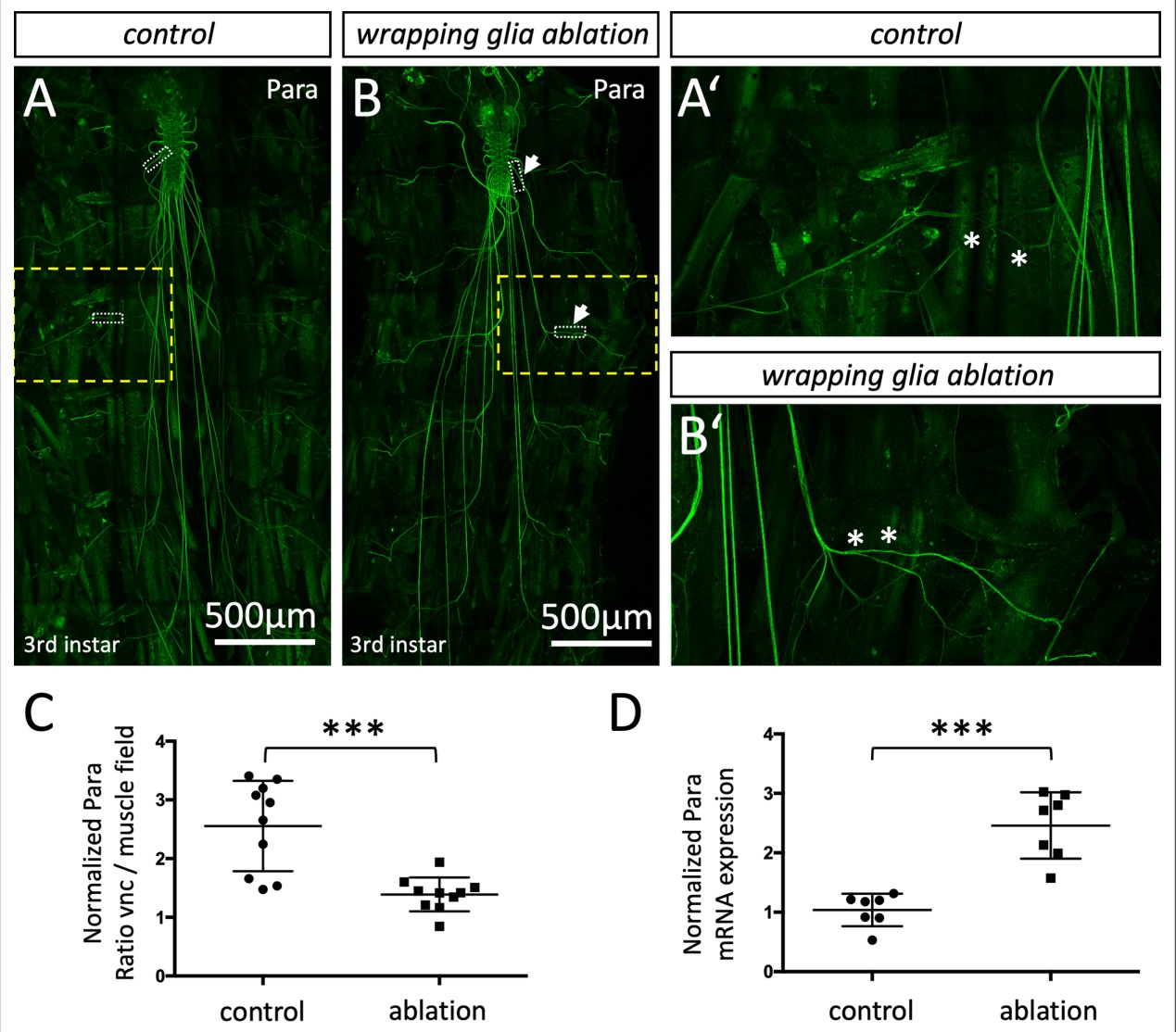

**Figure 8.** Localization of the voltage-gated sodium channel depends on glia. (**A,A'**) Third instar larval filet preparation with the genotype [*nrv2-Gal4, UAS-CD8-GFP; R90C03-Gal80*] showing the localization of Paralytic (Para) as detected using the anti-Para antibody in a control larva. (**B,B'**) Third instar larval filet preparation with the genotype [*nrv2-Gal4, UAS-hid; R90C03-Gal80*] showing the localization of Para as detected using the anti-Para antibody in a wrapping glia ablated larva. The white dashed boxes were used for quantification of Para fluorescence intensity in the CNS/PNS transition zone in relation to its expression in the muscle field area. The yellow boxed areas are shown in higher magnification (**A',B'**). Note the increased localization of Para along the peripheral nerve at the level of the muscle field (asterisks). Scale bars are as indicated. (**C**) Quantification of Para fluorescence intensity in the CNS/PNS transition area and the muscle field area in control and wrapping glia ablated larvae (n=10 larval filets, 3 nerves/filet). To exclude a possible influence seen in individual animals, the average fluorescence intensities along nerves of each individual were compared. Note, Para distributes more evenly along the axon in the absence of wrapping glia (p=0,0003; Mann-Whitney U-test). (**D**) Quantification of *para* mRNA expression using qRT-PCR in control and wrapping glia ablated larvae (n=7, with 15–20 brains each). *para* ct-values were normalized to ct-values of control gene, *RPL32*. Note, the significant increase in *para* mRNA expression upon wrapping glia ablation (p=0,0006, Mann-Whitney U-test). Scale bars are as indicated.

The online version of this article includes the following figure supplement(s) for figure 8:

**Figure supplement 1.** Glia ablation does not affect the localization of the voltage-gated potassium channel Shal.

**Figure supplement 2.** Ablation of central ensheathing glia does not affect positioning of Paralytic (Para) at the axon initial segment (AIS).

## Discussion

In the vertebrate nervous system, saltatory conductance allows very fast spreading of information. This requires localized distribution of voltage-gated ion channels and concomitantly, the formation of the myelin sheath. The evolution of this complex structure is unclear. Here, we report glial-dependent

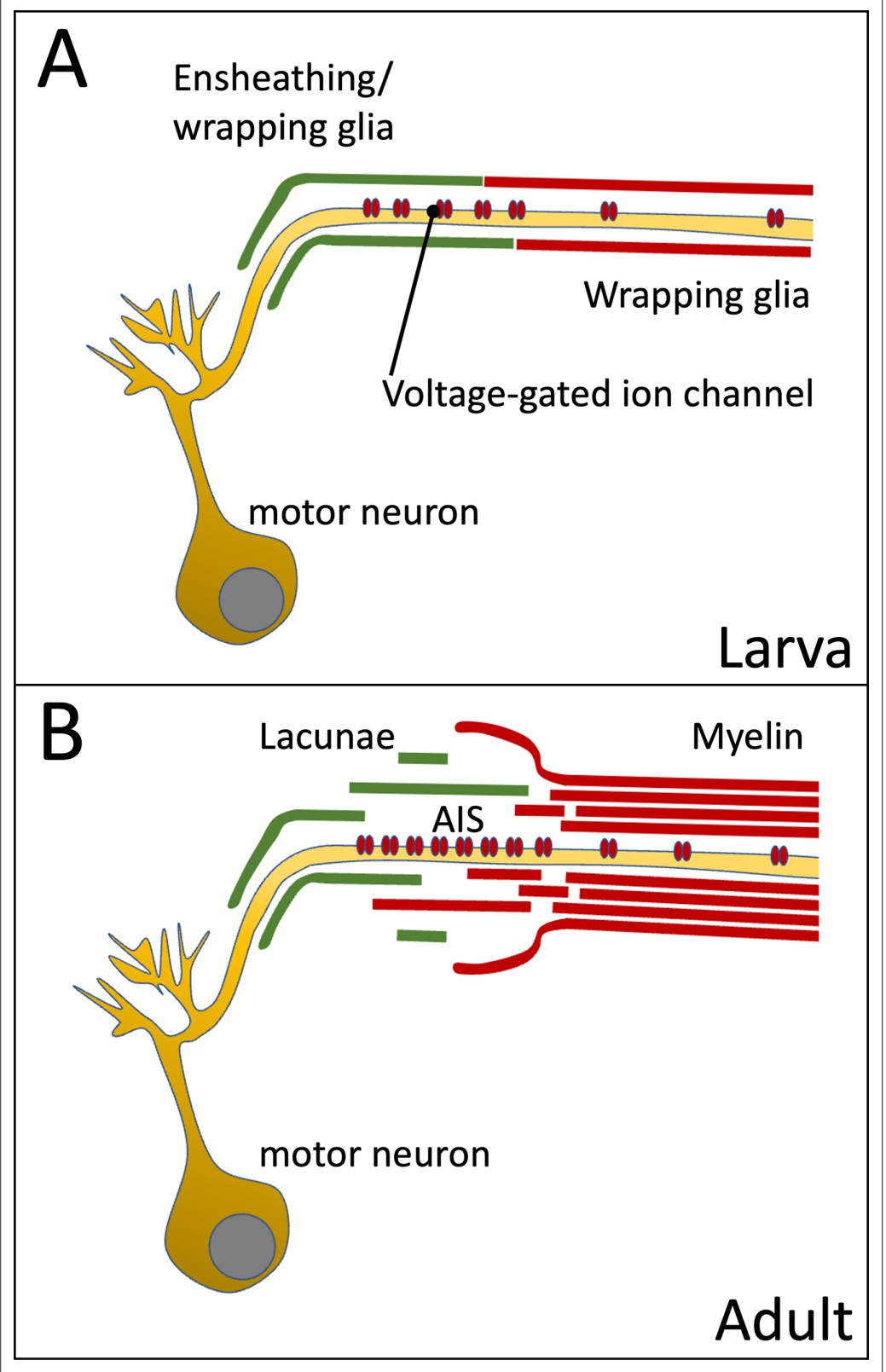

**Figure 9.** Organization of the axon initial segment (AIS) in *Drosophila* motor axons. Voltage-gated sodium channels are preferentially positioned at the AIS of the motor axon. (**A**) In the larval nervous system positioning is mediated by the peripheral wrapping glia. (**B**) In adults these cells form myelin-like structures, which fray out in the lacunae which represent a reservoir possibly needed for ion homeostasis during sustained action potential generation.

localization of voltage-gated ion channels at an AIS-like domain of peripheral *Drosophila* larval motor axons. As more channels accumulate in adults, a lacunar system and adjacent myelin-like structures are formed by central tract glia and peripheral wrapping glia.

In myelinated axons of vertebrates, voltage-gated Na$^+$ and K$^+$ channels are clustered at the AIS and the nodes of Ranvier (*Amor et al., 2014*; *Freeman et al., 2016*; *Nelson and Jenkins, 2017*). In invertebrate neurons, the AIS corresponds to the spike initiation zone located distal to the soma and distal to the dendrite branching point. Such segments were found in *Caenorhabditis elegans* (*Eichel et al., 2022*) and have been previously postulated for *Drosophila* neurons due to the localization of a giant ankyrin, which in all systems appears to be an important scaffolding protein at the AIS, as well as the presence of voltage-gated ion channels (*Dubessy et al., 2019*; *Freeman et al., 2015*; *Jegla et al., 2016*; *Ravenscroft et al., 2020*; *Trunova et al., 2011*). Moreover, recent modeling approaches at the example of the pioneering aCC motor neuron predicted the localization of voltage-gated ion channels at the CNS/PNS boundary (*Günay et al., 2015*), which very well matches the localization of the voltage-gated ion channels Para and Shal, as reported here. Interestingly, in *Drosophila para* mRNA expression as well as Para protein localization depend on the presence of peripheral wrapping glia. In glia ablated nerves, Para expression is increased and decorates the entire axonal membrane. This loss of a clustered distribution may contribute to the pronounced reduction in axonal conductance velocity noted earlier in such glia ablated animals (*Kottmeier et al., 2020*). In addition, we found an increased *para* mRNA expression. How glial cells control Para localization and how this is then transduced to an increased expression of *para* remains to be further studied. Since alterations in glial differentiation caused by manipulation of FGF-receptor signaling specifically in peripheral wrapping glia does not cause a change in Para expression or localization (*Figure 8—figure supplement 2F–H*), proteins secreted by wrapping glia might be needed for the correct positioning of voltage-gated ion channels (*Yuan and Ganetzky, 1999*).

In the adult nervous system, the AIS-like domain is embedded in glial lacunar regions formed by wrapping glial cell processes. The increased expression of Para within the AIS-like segments of adult brains is expected to generate strong ephaptic coupling forces (*Rey et al., 2022*; *Rey et al., 2021*). These are caused by ion flux through open channels which generate an electric field that is able to influence the gating of ion channels in closely neighboring axons (*Arvanitaki, 1942*; *Krnjevic, 1986*; *Rasminsky, 1980*). Ephaptic coupling helps to synchronize firing axons (*Anastassiou and Koch, 2015*; *Anastassiou et al., 2011*; *Han et al., 2018*; *Shneider and Pekker, 2015*), but is also detrimental to the precision of neuronal signaling in closely apposed axons (*Arvanitaki, 1942*; *Kottmeier et al., 2020*).

Ephaptic coupling is counteracted by the glial lacunar system, that spatially separates axons and adds more levels of wrapping. Furthermore, it was postulated that the lacunar system provides a large extracellular ion reservoir (*Wigglesworth, 1960*). Given the tight apposition of axonal and glial membranes with most parts of the nerve, which is in the range of 20 nm, only a very small interstitial fluid volume is normally present. Thus, action potential generation would deplete sodium and potassium ions very fast, and would prevent sustained neuronal activity. The development of lacunar structures might therefore provide sufficient amount of ion and at the same time physically separates axons to reduce the likelihood of ephaptic coupling. It will be interesting to test this hypothesis in the future.

Close to the lacunar structures we detected myelin-like structures. It appears that the glial processes that form the lacunae collapse to form compact myelin-like membrane sheets. Interestingly, myelin-like structures are not formed at the lateral borders of the lacunae but rather form at its distal end. This indicates that insulation is likely not a key function of the myelin-like structures, but rather these structures originate as a consequence of the collapsed lacunar system. Concomitant with the occurrence of the myelin-like differentiations we note a decrease in the Para ion channel density. At the same time, the need for a large ion reservoir decreases, favoring the formation of myelin-like structures.

A hallmark of vertebrate myelin is the spiral growth of the insulating glial membrane. This is generally not observed in large fly nerves where glial membrane sheets rather fold back than spirally grow around a single axon. Compared to myelinated vertebrate axons, this provides the disadvantage that axons are not entirely insulated. However, spiral growth can be seen in small nerves where less extensive wrapping is noted. An additional unique feature of vertebrate myelin is its compact organization which is mediated by the MBP (*Nave and Werner, 2021*). In contrast to vertebrate myelin where extra- and intercellular space is removed, fly myelin-like structures only show a compaction of the extracellular space, which is expected to increase resistance as the number of freely moving ions is diminished. A fully compact myelin state would require MBP-like proteins which have not been identified in the fly genome.

In conclusion, the evolution of myelin appears reflected in the different developmental stages of *Drosophila*. First, voltage-gated ion channels are clustered at the AIS with the help of *Drosophila* glia. Second, upon increased expression of such ion channels in the adult nervous system, an ion reservoir might be formed by the lacunar system. The collapse of glial processes in the non-lacunar regions then provides the basis of myelin formation. In the future, it will be interesting to identify glial-derived signals that ensure channel positioning and determine how neuronal signaling adjusts channel expression and triggers formation of myelin.

## Materials and methods

### *Drosophila* genetics

All fly stocks were raised and kept at room temperature (RT) on standard *Drosophila* food. All crosses were raised at 25°C.

To determine temperature sensitivity, five 3-day-old male and female flies were transferred to an empty vial with a foam plug. The vials were incubated in a water bath at 42°C for 1 min and then placed at RT. Flies were monitored every 15 s for 5 min. At least 100 males and females for each genotype were tested.

For MCFO experiments early, white pupae were collected, put in a fresh vial and heat shocked at 37°C for 1 hr. Pupae were placed back to 25°C and dissected a few days after hatching.

To generate $para^{mCherry}$ flies we employed the MiMIC insertion strain $para^{Mi8578}$ generated by the Bellen lab and we injected pBS-KS-attB1-2-PT-SA-SD-0-mCherry (DGRC Stock 1299; https://dgrc.bio.indiana.edu//stock/1299; RRID:DGRC_1299; *Venken et al., 2011*) into embryos with the following genotype: *y w Φ31/para^{Mi08578}*. Following crosses to *FM7c, y w* flies were tested by PCR to identify successful insertion events.

To generate $para^{Apex2}$ flies, we first removed mCherry encoding sequences from pBS-KS-attB1-2-PT-SA-SD-0-mCherry (DGRC#1299) using restriction enzymes and then inserted the *apex2* coding sequence (Addgene #49386, using the primers AAGGATCCGGAAAGTCTTACCCAACTGT and AAGGATCCGGCATCAGCAAACCCAAG). pBS-KS-attB1-2-PT-SA-SD-0-Apex2 was used to establish a $para^{Apex2}$ as described above. Flies were tested via single-fly PCR. To generate UAS-Myr-Flag-Apex2 flies, we cloned Apex2 using the primers CACCgactacaaggatgacgacgataa and cagggtcaggcgctcc into pUAST_Myr_rfA_attB, which was then inserted into the landing site 86Fb using established protocols (*Bischof et al., 2007*).

### Western blot analysis

Ten adult fly heads were homogenized in 50 µl RIPA buffer on ice. They were centrifuged at 4°C for 20 min at 13,000 rpm. The supernatant was mixed with 5× reducing Lämmli buffer and incubated for 5 min at 65°C. Fifteen µl of the samples were separated to an 8% SDS-gel and subsequently blotted onto a PVDF membrane (Amersham Hybond-P PVDF Membrane, GE Healthcare). Anti-Para antibodies were generated against the following N-terminal sequence (CAEHEKQKELERKRAEGE), affinity purified, and were used in a 1/1000 dilution. Experiments were repeated three times.

### Cell culture

Primary neural cell culture was preformed as described (*Prokop et al., 2012*). In brief, three to five stage 11 embryos were collected, chemically dechorionated, and homogenized in 100 µl dispersion medium. Following sedimentation for 5 min at 600 × *g*, cells were resuspended in 30 µl culture medium and applied to a glass bottom chamber (MatTek), sealed with a ConA-coated coverslip. Cultures were grown for 5–7 days. Experiments were repeated three times.

### qPCR

RNA was isolated from dissected larval brains using the RNeasy mini kit (QIAGEN) and cDNA was synthesized using Quantitect Reverse Transcription Kit (QIAGEN) according to the manufacturer's instructions. qPCR for all samples was performed using a Taqman gene expression assay (Life Technologies) in a StepOne Real-Time PCR System (Thermo Fisher, para: Dm01813740_m1, RPL32: Dm02151827_g1). RPL32 was used as a housekeeping gene. Expression levels of Para were normalized to RPL32.

### Immunohistochemistry

Larval filets: L3 wandering larvae were collected in PBS on ice. Larvae were placed on a silicon pad and attached with two needles at both ends, with the dorsal side facing up. They were cut with a

fine scissor at the posterior end. Following opening with a long cut from the posterior to the anterior end the tissue was stretched and attached to the silicon pad with additional four to six needles. Gut, fat body, and trachea were removed. Adult brains: Adult flies were anesthetized with $CO_2$ and were dipped into 70% ethanol. The head capsule was cut open with fine scissors and the tissue surrounding the brain removed with forceps. Legs and wings were cut off and the thorax opened at the dorsal side. The ventral nerve cord was carefully freed from the tissue. For fixation, dissected samples were either covered for 3 min with Bouin's solution or for 20 min with 4% PFA in PBS. Following washing with PBT samples were incubated for 1 hr in 10% goat serum in PBT. Primary antibody incubation was at 4°C followed. The following antibodies were used: anti-Para N-term, this study; anti-dsRed (Takara), anti-GFP (Abcam, Invitrogen), anti-Rumpel (*Yildirim et al., 2022*), anti-Repo (Hybridoma bank), rabbit α-V5 (1:500, Sigma-Aldrich), mouse α-HA (1:1000, Covance), rat α-Flag (1:200, Novus Biologicals). The appropriate secondary antibodies (Thermo Fisher) were incubated for 3 hr at RT. The tissues were covered with Vectashield mounting solution (Vector Laboratories) and stored at 4°C until imaging using an LSM880 Airyscan microscope, or a Elyra 7 microscope (Carl Zeiss AG Elyra 7 imaging, lateral resolution 80 nm with a voxel size of 30 nm × 30 nm × 100 nm). All stainings were repeated >5 times.

## High-pressure freezing

Three-week-old female flies were used with head, legs, and tip of abdomen removed. Following fixation in 4% formaldehyde (FA) in 0.1 M PHEM in a mild vacuum (–200 mbar), at RT for 45 min and three washes in 0.1 PHEM, the tissue was embedded in 3% low melting agarose for vibratome sectioning (Leica, VTS1200S). Samples were cut in PBS into 200 µm thick cross sections with 1 mm/s, 1.25 mm amplitude, and were placed into lecithin-coated 6 mm planchettes, filled with 20% PVP in 0.1 M PHEM and high-pressure frozen (Leica, HPM100). Seven specimens were sectioned. Freeze substitution was performed in 1 %OsO$_4$, 0.2% glutaraldehyde, 3% water in acetone at –90°C and stepwise dehydrated over 3 days. Samples were embedded in mixtures of acetone and epon.

## DAB staining and electron microscopy

Flies were injected with 4% FA in 0.1 M HEPES buffer and fixed at RT for 45 min. Following washes and incubation in 20 mM glycine in 0.1 M HEPES, samples were incubated in 0.05% DAB in 0.1 M HEPES at RT for 40 min; 0.03% $H_2O_2$ was added and the reaction was stopped after 5–10 min. The tissue was then fixed in 4% FA and 0.2% glutaraldehyde in 0.1 M HEPES at RT for 3 hr. After three times rinsing the tissue was fixed in 4% FA at RT overnight. The FA was replaced by 2% $OsO_4$ in 0.1 M HEPES for 1 hr on ice (dark). Uranyl acetate staining was performed en bloque using a 2% solution in $H_2O$ for 30 min (dark). Following an EtOH series (50%, 70%, 80%, 90%, and 96%) on ice for 3 min each step, final dehydration was done at RT with 2×100% EtOH for 15 min and two times propylene oxide for 30 min. Grids of high-pressure frozen samples were additionally counterstained with uranyl acetate and lead citrate. Following slow epon infiltration specimens were embedded in flat molds and polymerized at 60°C for 2 days.

Six specimens from three different fixation experiments were sectioned. Ultrathin sections were cut using a 35° ultra knife (Diatome) and collected in formvar-coated one slot copper grids. For imaging a Zeiss TEM 900 at 80 kV in combination with a Morada camera (EMSIS, Münster, Germany) operated by the software iTEM. Image processing was done using Adobe Photoshop and Fiji. Ultrathin sections of high-pressure frozen samples were examined at a Tecnai 12 biotwin (Thermo Fisher Scientific) and imaged with a 2K CCD veleta camera (EMSIS, Münster, Germany).

To plot the Para distribution across the axonal surface, an axon was serially sectioned (≈70 nm section thickness) and imaged. The images were cropped to the size of the axon in Fiji and aligned using Affinity Photo (software version 1.10.5.1342). The rotated/aligned images were loaded into Fiji and the segmented line tool was used to create ROIs on top of the axonal membrane in every section. The ROI was set to have the same starting point for the measurement. Para$^{Apex2}$ staining intensity was measured along the circumference of an axon on 16 sequentially sectioned EM images. The relative gray values were binned by a factor of 100. This was then interpolated in 3D. We used biharmonic spline interpolation from the MATLAB Curve Fitting Toolbox (software version 9.13.0.2105380 (R2022b) Update 2) to generate a surface plot.

```
Script:
x=ParaE(:,1);
```

```
y=ParaE(:,3);
z=ParaE(:,2);
xlin = linspace(min(x), max(x), 100);
ylin = linspace(min(y), max(y), 100);
[X,Y]=meshgrid(xlin, ylin);
% Z=griddata(x,y,z,X,Y,'natural');
% Z=griddata(x,y,z,X,Y,'cubic');
Z=griddata(x,y,z,X,Y,'v4');
mesh(X,Y,Z)
axis tight; hold on
plot3(x,y,z,'.','MarkerSize',15)
```

## Acknowledgements

We are grateful to all our colleagues for many discussions and P Deing, K Krukkert, K Mildner, and E Naffin for excellent technical assistance. T Zobel for help using the Elyra 7 microscope. B Zalc and KA. Nave for critical reading of the manuscript and many thoughtful suggestions. This work was supported by the Deutsche Forschungsgemeinschaft through funds to CK. (SFB 1348, B5, Kl 588/29).

## Additional information

### Funding

| Funder | Grant reference number | Author |
| --- | --- | --- |
| Deutsche Forschungsgemeinschaft | SFB 1348 B5 | Christian Klämbt |
| Deutsche Forschungsgemeinschaft | Kl 588 / 29 | Christian Klämbt |

The funders had no role in study design, data collection and interpretation, or the decision to submit the work for publication.

### Author contributions

Simone Rey, Henrike Ohm, Conceptualization, Formal analysis, Investigation, Methodology, Writing – review and editing; Frederieke Moschref, Investigation, Methodology, Writing – review and editing; Dagmar Zeuschner, Methodology, Writing – review and editing; Marit Praetz, Formal analysis, Visualization; Christian Klämbt, Conceptualization, Supervision, Writing - original draft, Writing – review and editing

### Author ORCIDs

Simone Rey http://orcid.org/0000-0002-0130-6744
Henrike Ohm http://orcid.org/0009-0000-9156-3527
Dagmar Zeuschner http://orcid.org/0000-0002-6712-0192
Christian Klämbt http://orcid.org/0000-0002-6349-5800

### Decision letter and Author response

Decision letter https://doi.org/10.7554/eLife.85752.sa1
Author response https://doi.org/10.7554/eLife.85752.sa2

## Additional files

### Supplementary files
• MDAR checklist

## Data availability

All imaging and source data are available through: https://doi.org/10.57860/min_prj_000008. All *Drosophila* strains reported are available upon request to CK.

The following dataset was generated:

| Author(s) | Year | Dataset title | Dataset URL | Database and Identifier |
|---|---|---|---|---|
| Ohm H, Rey S | 2023 | Raw image data of all publication figures | https://doi.org/10.57860/min_prj_000008 | OMERO, 10.57860/min_prj_000008 |

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

# Appendix 1

## Appendix 1—key resources table

| Reagent type (species) or resource | Designation | Source or reference | Identifiers | Additional information |
|---|---|---|---|---|
| Genetic reagent (*Drosophila melanogaster*) | y[1] w[*] Mi{MIC}MI08578a Mi{MIC}MI08578b | Bloomington Drosophila Stock Center | BDSC51087 | |
| Genetic reagent (*Drosophila melanogaster*) | y[1] w[*] Mi{FlpStop}para[MI08578-FlpStop.D]/FM7c | Bloomington Drosophila Stock Center | BDSC67680 | |
| Genetic reagent (*Drosophila melanogaster*) | Para-mCherry | This study | | *Figures 1, 2* |
| Genetic reagent (*Drosophila melanogaster*) | Para-Apex | This study | | *Figures 1, 5* |
| Genetic reagent (*Drosophila melanogaster*) | Para-FlpTag GFP/Fm7i | *Fendl et al., 2020* | | |
| Genetic reagent (*Drosophila melanogaster*) | y[1] w[*]; Mi{PT-GFSTF.1}Shal[MI00446-GFSTF.1] | Bloomington Drosophila Stock Center | BDSC60149 | |
| Genetic reagent (*Drosophila melanogaster*) | y[1] w[*] Mi{PT-GFSTF.2}Sh[MI10885-GFSTF.2]/FM7j, B[1] | Bloomington Drosophila Stock Center | BDSC59423 | |
| Genetic reagent (*Drosophila melanogaster*) | y[1] w[*]; Mi{PT-GFSTF.1}Shab[MI00848-GFSTF.1]/TM6C, Sb[1] Tb[1] | Bloomington Drosophila Stock Center | BDSC60514 | |
| Genetic reagent (*Drosophila melanogaster*) | y[1] sc[*] v[1] sev[21]; P{y[+t7.7] v[+t1.8]=VALIUM20-mCherry}attP2 | Bloomington Drosophila Stock Center | BDSC35785 | |
| Genetic reagent (*Drosophila melanogaster*) | UAS-CD8GFP II; R90C03Gal80 III | *Kottmeier et al., 2020* | | |
| Genetic reagent (*Drosophila melanogaster*) | UAS-Hid/CyOw; R90C03Gal80 III | *Kottmeier et al., 2020* | | |
| Genetic reagent (*Drosophila melanogaster*) | UAS-lacZ NLS II | Y. Hirmoi | | |
| Genetic reagent (*Drosophila melanogaster*) | UAS-lambda-Htl | *Gisselbrecht et al., 1996* | | |
| Genetic reagent (*Drosophila melanogaster*) | UAS-htl$^{DN}$ II | Bloomington Drosophila Stock Center | BDSC5366 | |
| Genetic reagent (*Drosophila melanogaster*) | UAS-CD8GFP II | Bloomington Drosophila Stock Center | BDSC5137 | |
| Genetic reagent (*Drosophila melanogaster*) | UAS-CD8mCherry II | Bloomington Drosophila Stock Center | BDSC 27391 | |
| Genetic reagent (*Drosophila melanogaster*) | UAS-Hid II | *Igaki et al., 2000* | | |
| Genetic reagent (*Drosophila melanogaster*) | UAS-Flp | Bloomington Drosophila Stock Center | BDSC 4539 | |
| Genetic reagent (*Drosophila melanogaster*) | UAS-Myr-Flag-APEX2-NES[86Fb] III | This study | | *Figures 6, 7* |
| Genetic reagent (*Drosophila melanogaster*) | pBPhsFLP2::pEST/I;; UAS HA, FLAG, V5, OLLAS/III | *Nern et al., 2015* | | |

*Appendix 1 Continued on next page*

*Appendix 1 Continued*

| Reagent type (species) or resource | Designation | Source or reference | Identifiers | Additional information |
|---|---|---|---|---|
| Genetic reagent (*Drosophila melanogaster*) | ChAT-Gal4 | **Salvaterra and Kitamoto, 2001** | | |
| Genetic reagent (*Drosophila melanogaster*) | OK371-Gal4 | **Mahr and Aberle, 2006** | | |
| Genetic reagent (*Drosophila melanogaster*) | GMR94G06Gal4 III | Bloomington Drosophila Stock Center | BDSC40701 | |
| Genetic reagent (*Drosophila melanogaster*) | Nrv2Gal4 II | **Sun et al., 1999** | | |
| Genetic reagent (*Drosophila melanogaster*) | Nrv2Gal4 II; R90C03Gal80 III | **Kottmeier et al., 2020** | | |
| Genetic reagent (*Drosophila melanogaster*) | Nrv2Gal4 II; R90C03Gal80, UAS-CD8Cherry III | **Kottmeier et al., 2020** | | |
| Genetic reagent (*Drosophila melanogaster*) | GMR83E12_AD II; Repo4.3_DBD III | **Bittern et al., 2021** | | |
| Genetic reagent (*Drosophila melanogaster*) | GMR75H03-Gal4 III | Bloomington Drosophila Stock Center | BDSC39908 | |
| Genetic reagent (*Drosophila melanogaster*) | ppkGal4, AUS-tdTomato III | **Herzmann et al., 2017** | | |
| Genetic reagent (*Drosophila melanogaster*) | lexAop-Flp III | Bloomington Drosophila Stock Center | BDSC55819 | |
| Genetic reagent (*Drosophila melanogaster*) | w[*]; TI{2A-lexA::GAD}VGlut[2A-lexA]/CyO | Bloomington Drosophila Stock Center | BDSC84442 | |
| Genetic reagent (*Drosophila melanogaster*) | Para$^{ST76}$ | Bloomington Drosophila Stock Center | BDSC26701 | |
| Genetic reagent (*Drosophila melanogaster*) | Oregon R | Bloomington Drosophila Stock Center | BDSC5 | |
| Antibody | Anti-Para N-term (rabbit, polyclonal) | This study | | IF (1:1000), WB (1:1000) **Figures 1, 2, 3** |
| Antibody | Anti-dsRed (rabbit, polyclonal) | Takara | Cat#632496 RRID:AB_10013483 | IF (1:1000) |
| Antibody | Anti-GFP (chicken, polyclonal) | Abcam | Cat#ab13970 RRID:AB_300798 | IF(1:500) |
| Antibody | Anti-GFP (rabbit, polyclonal) | Invitrogen | Cat#**A-11122** RRID:AB_221569 | IF(1:1000) |
| Antibody | Anti-Rumpel (rabbit, polyclonal) | **Yildirim et al., 2022** | | IF(1:1000) |
| Antibody | Anti-Repo (mouse, monoclonal) | Hybridoma Bank | Cat#8D12 RRID: AB_528448 | IF(1:5) |
| Antibody | Anti-V5 (rabbit, polyclonal) | Sigma-Aldrich | Cat#V8137-.2MG RRID:AB_261889 | IF(1:500) |
| Antibody | Anti-HA (mouse, monoclonal) | Covance BioLegend | Cat#MMS-101R RRID:AB_291262 | IF(1:1000) |
| Antibody | Anti-Flag (rat, monoclonal) | Novus Biologicals | Cat#NBP1-06712SS RRID:AB_162598 | IF(1:200) |
| Antibody | FluoTag-X4 anti-GFP (Alpaca, monoclonal) | NanoTag Biotechnologies | Cat#N0304 RRID:AB_2905516 | IF(1:500) |
| Antibody | FluoTag-X4 anti-RFP (Alpaca, monoclonal) | NanoTag Biotechnologies | Cat#N0404 RRID:AB_2744638 | IF(1:500) |

*Appendix 1 Continued on next page*

*Appendix 1 Continued*

| Reagent type (species) or resource | Designation | Source or reference | Identifiers | Additional information |
|---|---|---|---|---|
| Antibody | Anti-rabbit Alexa 488 (goat, polyclonal) | Thermo Fisher | Cat#A-11008 RRID:AB_143165 | IF(1:1000) |
| Antibody | Anti-rabbit Alexa 568 (goat, polyclonal) | Thermo Fisher | Cat#A-11011 RRID:AB_143157 | IF(1:1000) |
| Antibody | Anti-chicken Alexa 488 (goat, polyclonal) | Thermo Fisher | Cat#A-11039, RRID:AB_2534096 | IF(1:1000) |
| Antibody | Anti-mouse Alexa 488 (goat, polyclonal) | Thermo Fisher | Cat#A-11001, RRID:AB_2534069 | IF(1:1000) |
| Antibody | Anti-HRP Alexa 647 (goat, polyclonal) | Thermo Fisher | Cat#123-605-021, RRID:AB_2338967 | IF(1:500) |
| Antibody | Anti-rabbit HRP (goat, polyclonal) | Invitrogen | Cat#31460 RRID:AB_228341 | WB(1:5000) |
| Recombinant DNA reagent | pBS-KS-attB1-2-PT-SA-SD-0-mCherry | *Drosophila* Genome Research Centre | DGRC#1299 | |
| Recombinant DNA reagent | pBS-KS-attB1-2-PT-SA-SD-0-Apex2 | This study | | Generation of transgenic fly |
| Sequence-based reagent | BamH1 Apex2 fwd | This study | PCR primer | AAGGATCCGGAAAGTCTTACCCAACTGT |
| Sequence-based reagent | BamH1 Apex2 rev | This study | PCR primer | AAGGATCCGGCATCAGCAAACCCAAG |
| Sequence-based reagent | MiLF | *Venken et al., 2011* | PCR primer | GCGTAAGCTACCTTAATCTCAAGAAGAG |
| Sequence-based reagent | MiLR | *Venken et al., 2011* | PCR primer | CGCGGCGTAATGTGATTTACTATCATAC |
| Sequence-based reagent | mCherry-Seq fwd | | PCR primer | ACGGCGAGTTCATCTACAAG |
| Sequence-based reagent | mCherry-Seq rev | | PCR primer | TTCAGCCTCTGCTTGATCTC |
| Sequence-based reagent | Apex253_rev1 | This study | PCR primer | AGCTCAAAATAGGGAACTCCG |
| Sequence-based reagent | Apex286_fwd1 | This study | PCR primer | TACCAGTTGGCTGGCGTTGTT |
| Sequence-based reagent | Para qPCR Primer | Thermo Fisher Scientific | Cat#4331182 Dm01813740_m1 | |
| Sequence-based reagent | RPL32 qPCR Primer | Thermo Fisher Scientific | Cat#4331182 Dm02151827_g1 | |
| Commercial assay or kit | RNeasy Kit | QIAGEN | Cat#74104 | |
| Commercial assay or kit | Quantitect Reverse Transcription Kit | QIAGEN | Cat#205313 | |
| Commercial assay or kit | Taqman gene expression assay, Universal Master Mix II, with UNG | Thermo Fisher Scientific | Cat#4440038 | |
| Software algorithm | GraphPad PRISM | GraphPad Software, USA | Version 6.0 | |
| Software algorithm | Fiji | https://imagej.net/software/fiji/ | | |
| Software algorithm | ZEN Software | Zeiss | Black version | |
| Software algorithm | Affinity Photo | Serif (Europe) | | |
| Software algorithm | MATLAB | The MathWorks, Inc | | |
| Software algorithm | Photoshop CS6 | Adobe | | |

