## [Editor Report]

Here, the authors characterize axon-initial segment-like structures in *Drosophila* using a variety of approaches spanning molecular genetic, confocal and super-resolution imaging, and ultrastructure. These important findings advance our understanding of how myelin evolved with potentially early roles in organizing ion channel clustering in axons. The convincing evidence supporting the conclusions includes advanced imaging combined with molecular genetic approaches, which will be of significant interest to neuroscientists and cell biologists.

---

## [Decision Letter]

**Decision letter after peer review:**

Thank you for submitting your article "Glial-dependent clustering of voltage-gated ion channels in *Drosophila* precedes myelin formation" for consideration by *eLife*. Your article has been reviewed by 2 peer reviewers, and the evaluation has been overseen by a Reviewing Editor and Claude Desplan as the Senior Editor. The reviewers have opted to remain anonymous.

Essential revisions:

There were a number of concerns raised regarding the strength of the data needed to justify the conclusions made. In particular, the points below and more details in the reviews should be carefully considered:

1) There are significant areas in which the conclusions stated by the authors appear to border on speculation of attractive possibilities rather than derive from the actual experimental data presented in the manuscript.

2) Quantitatively define the percentage of axons within the more heavily myelinated regions that are actually enveloped by myelin (spiral or otherwise). For the Apex2 experiments, quantify the longitudinal distribution of para along more than one axon.

*Reviewer #1 (Recommendations for the authors):*

This study characterizes the localization of the lone voltage-gated Na channel in *Drosophila* Para in motor and sensory neurons. Like previous studies, the authors identify a enrichment of Na (and importantly the K^+^ channel Shal) in axon initial segment-like areas in motor neuron axons, and show that this structure is not apparent in axons of sensory neurons. Upon ablation of wrapping glia in the periphery, the authors find this AIS-like organization of Para is lost. Finally, beautiful EM analyses of peripheral nerves suggest an intriguing area devoid of glia around AIS-like structures, and some evidence for myelin-like structures along the distal axon. The author propose several interesting ideas for how these structures might be involved in AP signaling and as evolutionary precursors to conventional myelination and saltatory conductance in vertebrates.

Clearly, the evolution of myelination, and how glia contribute to neuronal firing in systems without classically accepted myelination and saltatory conductance are important questions. Although much of the Para clustering in AIS-like domains and regular densities along motor axons have been described in previous studies, the ultrastructural analyses and dependence on wrapping glia might be important advances to the field. In particular, major strengths of this study are the detailed analysis of AIS-like Para clusters, spanning molecular genetic, confocal and super resolution imaging, and ultrastructural approaches and clear writing. However, these strengths are somewhat tempered by a lack of functional approaches to test the idea of a lacunar structure that promotes ion exchange at putative AIS regions as well as little mechanistic insight into how glia may specifically coordinate the formation of Para clusters in AIS-like regions. Clearly, glia are dispensable for the intermittent clustering of Para along distal axons (and sensory axons), and the extent to which the glial sheet structures observed by EM actually function as myelin-like insulation is not clear. The ideas presented in this study about how glia might promote AIS organization and function are of high interest and potential importance, but the data as presented appear preliminary and do not yet provide a compelling foundation to fully support the conclusions drawn.

1. As the authors note, endogenously tagged para alleles were previously generated and characterized (Ravenscroft et al., 2020). The authors state they also generated their own tagged alleles, but it's not clear in the results which alleles were made and whether the alleles from each group differ in any meaningful way. This should be clarified and explicitly mentioned. There also weren't significant details explained in the methods of how the new para.mCherry allele was generated in this study.

It should be noted that a GFP-tagged para allele exists from the 2020 study, which might be used to control for possible mCherry-mediated clustering or disruption of Para function. Indeed, it appears the mCherry tagging significantly disrupts para function from the paralytic assay shown in Figure 1B. Do the GFP and/or mCherry tagged para alleles from the Ravenscroft study behave similarly? That being said, the authors do use new anti-Para antibodies and the previously published conditional para.GFP allele to support their results on the mCherry allele.

Finally, credit should also be explicitly given to the Ravenscroft study for initially describing Para in AIS-like regions in MNs as well as their regular spacing along distal axons. This doesn't seem to be discussed or acknowledged in the current manuscript, in which Figure 1 and 2 in the current manuscript largely confirm this data.

2. Functional data would really strengthen this study to better understand how Para localization in AIS-like structures contributes to AP generation and conductance and the role of glia. Voltage imaging of motor neurons vs sensory neurons (which apparently lack Para/AIS-like structures) would be ideal, but it is understood this is technically very challenging. An important question is how the AP is propagated along motor neuron vs sensory axons given the apparently regular spacing of Para. Are Para channels similarly spaced every 0.6-0.8 μm in sensory neurons?

3. There are significant areas in which the conclusions stated by the authors appear to border on speculation of attractive possibilities rather than derive from the actual experimental data presented in the manuscript. For example, there is no functional data to support the speculation about the lacunar structures and whether they actually influence or stabilize ion exchange at Para/AIS clusters. In addition, it is not clear that glia directly control Para/AIS formation themselves, or rather whether a loss of these structures are a secondary consequence of chronic ablation of glia and subsequent developmental changes. While these ideas and others discussed by the authors are interesting to consider and may indeed be the case, they should be presented in the discussion and not as prominently in the abstract or results (or at least softened to acknowledge the limitations and lack of direct experimental data in some cases).

*Reviewer #2 (Recommendations for the authors):*

Overall, this data was interesting but greatly suffered from a lack of quantification and clarity. The authors should revisit their figure legends and make it more clear how many samples were assessed per experiment (not only number of animals, but number of axons when it comes to the TEM studies). I would suggest the following analyses to bolster their claims. Note that these analyses should be possible with the data that the authors already have:

By TEM:

1. What percentage of axons within the more heavily myelinated regions are actually enveloped by myelin (spiral or otherwise)? This should be correlated with axon diameter (e.g. axons smaller than 1 micron, none were myelinated, axons between 1-3 microns, half myelinated, etc).

2. For the Apex2 experiments, the authors should quantify the longitudinal distribution of para along more than one axon. If they have already, they should show an averaged trace (Figure 3F).

3. Figure 4I, is the interperiodic distance between myelin wraps the same or different than in vertebrates?

In addition, I ask for additional clarity relating to the Para/Shal clustering experiments. In their in vitro experiments, they were able to find Para clustering even without many glia present (page 7), suggesting that there are neuron-intrinsic mechanisms for clustering. How do the authors reconcile this with their in vivo analyses in figure 5? I will note that it would be helpful to have an HRP counterstain in Figure 5 so that the levels can be normalized to the nerve itself, which is difficult to see in the controls as it gets closer to the muscle.

Finally, I will reiterate that the figures were difficult to follow due to lack of labels throughout. The addition of genotypes and labeling related to antibodies used would be of great help.

---

## [Author Response]

Essential revisions:There were a number of concerns raised regarding the strength of the data needed to justify the conclusions made. In particular, the points below and more details in the reviews should be carefully considered:1) There are significant areas in which the conclusions stated by the authors appear to border on speculation of attractive possibilities rather than derive from the actual experimental data presented in the manuscript.

We carefully revised the manuscript and removed all speculations from the results part. In the discussion they are clearly labelled and put into the respective context.

2) Quantitatively define the percentage of axons within the more heavily myelinated regions that are actually enveloped by myelin (spiral or otherwise). For the Apex2 experiments, quantify the longitudinal distribution of para along more than one axon.

We have now quantified the myelinated regions in relation to the extent of the lacunae and gave numbers on the number of axons showing myelin. In addition, we carefully quantified the longitudinal distribution of Para along several axons. For this we generated new data allowing to study the distribution of Para clusters along a single defined motor axon (employing the FlpTag technique and a new Zeiss Elyra 7 microscope that we were able to test). These, as we think, amazing data provide quantitative information on the spacing of 100 Para clusters. These data are shown in a new Figure 3. Similarly, we quantified the distribution of Para^Apex2^ clusters on 8 longitudinal axons. In both cases we observed an average spacing of about 650 – 700 nm (also shown in the new Figure 3).

Reviewer #1 (Recommendations for the authors):This study characterizes the localization of the lone voltage-gated Na channel in *Drosophila* Para in motor and sensory neurons. Like previous studies, the authors identify a enrichment of Na (and importantly the K^+^ channel Shal) in axon initial segment-like areas in motor neuron axons, and show that this structure is not apparent in axons of sensory neurons. Upon ablation of wrapping glia in the periphery, the authors find this AIS-like organization of Para is lost. Finally, beautiful EM analyses of peripheral nerves suggest an intriguing area devoid of glia around AIS-like structures, and some evidence for myelin-like structures along the distal axon. The author propose several interesting ideas for how these structures might be involved in AP signaling and as evolutionary precursors to conventional myelination and saltatory conductance in vertebrates.Clearly, the evolution of myelination, and how glia contribute to neuronal firing in systems without classically accepted myelination and saltatory conductance are important questions. Although much of the Para clustering in AIS-like domains and regular densities along motor axons have been described in previous studies, the ultrastructural analyses and dependence on wrapping glia might be important advances to the field. In particular, major strengths of this study are the detailed analysis of AIS-like Para clusters, spanning molecular genetic, confocal and super resolution imaging, and ultrastructural approaches and clear writing. However, these strengths are somewhat tempered by a lack of functional approaches to test the idea of a lacunar structure that promotes ion exchange at putative AIS regions as well as little mechanistic insight into how glia may specifically coordinate the formation of Para clusters in AIS-like regions. Clearly, glia are dispensable for the intermittent clustering of Para along distal axons (and sensory axons), and the extent to which the glial sheet structures observed by EM actually function as myelin-like insulation is not clear. The ideas presented in this study about how glia might promote AIS organization and function are of high interest and potential importance, but the data as presented appear preliminary and do not yet provide a compelling foundation to fully support the conclusions drawn.1. As the authors note, endogenously tagged para alleles were previously generated and characterized (Ravenscroft et al., 2020). The authors state they also generated their own tagged alleles, but it's not clear in the results which alleles were made and whether the alleles from each group differ in any meaningful way. This should be clarified and explicitly mentioned. There also weren't significant details explained in the methods of how the new para.mCherry allele was generated in this study.

We had started the project already in 2018 and our Para^mCherry^ allele was thus generated before the Ravenscroft paper was published. We clarified this and added a section on Material and Methods detailing how we generated the Para^mCherry^ allele.

It should be noted that a GFP-tagged para allele exists from the 2020 study, which might be used to control for possible mCherry-mediated clustering or disruption of Para function. Indeed, it appears the mCherry tagging significantly disrupts para function from the paralytic assay shown in Figure 1B. Do the GFP and/or mCherry tagged para alleles from the Ravenscroft study behave similarly?

Interestingly, the Para^mCherry^ allele generated in our study behaves as a slightly weaker hypomorphic allele when compared to the Para^GFP^ allele generated by Ravenscroft (taking the data from the paper). Moreover, the Para^Apex2^ allele is an even weaker allele (our data, presented in the manuscript). The conditional para^flpTagGFP^ allele generated in the Borst lab, carries an GFP insertion at the same position as the other gene trap lines. Since all insertions were made using the same landing site, it appears that mCherry, GFP or Apex2 each have a specific but relatively weak effect on the stability of the protein.

That being said, the authors do use new anti-Para antibodies and the previously published conditional para.GFP allele to support their results on the mCherry allele.

We are aware of the fact that mCherry or GFP proteins could confer ectopic clustering of the tagged proteins. Therefore, we generated the Para specific antibodies, which also demonstrate that Para channels have distinct subcellular localization and show that the pattern observed using mCherry or GFP tagged Para corresponds to the wild typic situation.

Finally, credit should also be explicitly given to the Ravenscroft study for initially describing Para in AIS-like regions in MNs as well as their regular spacing along distal axons. This doesn't seem to be discussed or acknowledged in the current manuscript, in which Figure 1 and 2 in the current manuscript largely confirm this data.

In the revised version of the manuscript we made sure to properly acknowledge the study by Ravenscroft et al.

2. Functional data would really strengthen this study to better understand how Para localization in AIS-like structures contributes to AP generation and conductance and the role of glia. Voltage imaging of motor neurons vs sensory neurons (which apparently lack Para/AIS-like structures) would be ideal, but it is understood this is technically very challenging. An important question is how the AP is propagated along motor neuron vs sensory axons given the apparently regular spacing of Para. Are Para channels similarly spaced every 0.6-0.8 μm in sensory neurons?

We fully agree with this comment but we unfortunately failed to get Voltron-based imaging to work. To at least discuss possible differences between sensory- and motor- axons, we determined the distribution of Para channels in the two modalities. Although, Para is localized in an axon initial segment (now shown in the paper), the concentration and thus the staining intensity did not allow high resolution microscopy. The following is now added in addition with the new Figure 4:

“Having shown that in motor axons Para is concentrated in a clustered arrangement in an axon initial segment of motor axons we wondered whether similar distribution can be found in sensory axons. For this we expressed *flp* in multidendritic sensory neurons using the *pickpocket* Gal4 driver (*ppk-Gal4*). This allows labeling of the v’ada neurons (Figure 4A). Low levels of Para protein localize to the cell body and the distal shaft of the axon. Along the descending axon, Para localization increases only in some distance to the soma (Figure 4A,B,C). However, *para* expression in sensory neurons is not as strong as in motor axons which may correspond to the notion that sensory axons are usually smaller axons. The relatively low expression levels did not allow super-resolution imaging and thus, we could not address whether Para is found in a clustered organization along axons of *ppk* positive sensory neurons. Interestingly, however, within some of the v’ada dendrites, Para accumulates in distinct clusters (Figure 4A’,B’ arrows).”

Reviewer #2 (Recommendations for the authors):Overall, this data was interesting but greatly suffered from a lack of quantification and clarity. The authors should revisit their figure legends and make it more clear how many samples were assessed per experiment (not only number of animals, but number of axons when it comes to the TEM studies). I would suggest the following analyses to bolster their claims. Note that these analyses should be possible with the data that the authors already have:By TEM:1. What percentage of axons within the more heavily myelinated regions are actually enveloped by myelin (spiral or otherwise)? This should be correlated with axon diameter (e.g. axons smaller than 1 micron, none were myelinated, axons between 1-3 microns, half myelinated, etc).

We have now quantified formation of myelin-like structures. This however, is not as straightforward as we had initially anticipated and did require the generation of new series of TEM images. We defined “myelin-like” in *Drosophila* as a glial derived membrane stack comprising ≥4 closely stacked glial cell layers. We then quantified the occurrence of myelin in relation to the lacunar structures. For this we counted the number of myelin-like structures in serial sections (every 5 µm) along the distal – proximal axis. The distal most section that showed lacunar structures was defined as 0, which was then used to align all leg nerves. Firstly, we counted the number of myelin-like stacks per section. As this did not reflect the size of the myelin-like stack, we, secondly, counted the number of axons associated with different myelin-like stacks. Due to the high heterogeneity in myelin-like morphology, this might not be a precise measure to account for myelin-like distribution but in our view provide the best information on the distribution, which shows more myelin-like structures towards the distal end of the lacunae (new Figures: Figure 7—figure supplements 2 and 3).

2. For the Apex2 experiments, the authors should quantify the longitudinal distribution of para along more than one axon. If they have already, they should show an averaged trace (Figure 3F).

We have now quantified cluster formation in three different paradigms. First, we quantified the longitudinal distribution of Para^mCherry^ clusters in single labelled motor axons using super resolution imaging (using the new Elyra 7 microscope). Second, we quantified longitudinal distribution of Para^Apex2^ clusters in single labelled motor axons at the electron microscope level. Both analyses give very similar values as shown in the new Figure 4F-G.

Finally, we determined the circular distribution of DAB precipitates in ParaApex2 animals in 16 consecutive TEM cross sections as shown in the new Figure 5E-G.

Together this analysis indicates that Para channels are clustered along lines.

3. Figure 4I, is the interperiodic distance between myelin wraps the same or different than in vertebrates?

Since the organization of vertebrate and *Drosophila* myelin is not comparable we have not used these terms. We now measured the distance of the membrane stack distance and added:

“To further validate these findings we performed additional high pressure freezing of pre-fixed samples to optimize tissue preservation (Sosinsky et al., 2008). In such specimens compact stackings of thin glial membrane sheets can be detected, too (Figure 5J,K). Here spacing of different glial layers is 30 nm (Figure 5J,K) which is considerably more than the interperiodic spacing of 13 nm found in mouse peripheral myelin [Fledrich, 2018 #5161].”

In addition, I ask for additional clarity relating to the Para/Shal clustering experiments. In their in vitro experiments, they were able to find Para clustering even without many glia present (page 7), suggesting that there are neuron-intrinsic mechanisms for clustering. How do the authors reconcile this with their in vivo analyses in figure 5? I will note that it would be helpful to have an HRP counterstain in Figure 5 so that the levels can be normalized to the nerve itself, which is difficult to see in the controls as it gets closer to the muscle.

We apologize for the unclear description of the *Drosophila* cell culture experiments. Here, neuro(glio)blast cells are isolated from embryos which generate both neurons and glial cells.

Glial cell nuclei are labelled by anti-Repo staining. To normalize for differential expression levels along the nerve we have used Shal channel distribution which is not affected by glial ablation. Normalizing to HRP is problematic as anti-HRP antibodies recognize a glycol-epitope and in vertebrates, voltage-gated sodium channels are heavily glycosylated.

Finally, I will reiterate that the figures were difficult to follow due to lack of labels throughout. The addition of genotypes and labeling related to antibodies used would be of great help.

We are thankful for this comment and have added genotypes and staining information to all Figures. We are thankful for all the positive and constructive comments and are hoping that the manuscript can now be accepted for publication.